# Spatio-temporal controls of C-N-P dynamics across headwater catchments of a temperate agricultural region from public data analysis

Stella Guillemot[1,2], Ophelie Fovet[1], Chantal Gascuel-Odoux[1], Gérard Gruau[3], Antoine Casquin[1], Florence Curie[2], Camille Minaudo[4], Laurent Strohmenger[1], and Florentina Moatar[5,2]

[1]INRAE, AGROCAMPUS OUEST/INSTITUT AGRO, UMR SAS, 35000 Rennes, France
[2]Université de Tours, EA 6293 GéHCO, 37200 Tours, France
[3]OSUR, Geosciences Rennes, CNRS, Université Rennes 1, 35000 Rennes, France
[4]EPFL, Physics of Aquatic Systems Laboratory, 1015 Lausanne, Switzerland
[5]INRAE, RIVERLY, 69625 Villeurbanne, France

*Correspondence to*: Ophelie Fovet (ophelie.fovet@inrae.fr)

**Abstract.** Characterizing and understanding spatial variability in water quality for a variety of chemical elements is an issue for present and future water resource management. However, most studies of spatial variability in water quality focus on a single element and rarely consider headwater catchments. Moreover, they assess few catchments and focus on annual means without considering seasonal variations. To overcome these limitations, we studied spatial variability and seasonal variation in dissolved C, N, and P concentrations at the scale of an intensively farmed region of France (Brittany). We analyzed 185 headwater catchments (from 5-179 km²) for which 10-year time series of monthly concentrations and daily stream flow were available from public databases. We calculated interannual loads, concentration percentiles, and seasonal metrics for each element to assess their spatial patterns and correlations. We then performed rank correlation analyses between water quality, human pressures, and soil and climate features. Results show that nitrate ($NO_3$) concentrations increased with increasing agricultural pressures and base flow contribution; dissolved organic carbon (DOC) concentrations decreased with increasing rainfall, base flow contribution, and topography; and soluble reactive phosphorus (SRP) concentrations showed weaker positive correlations with diffuse and point sources, rainfall and topography. An opposite pattern was found between DOC and $NO_3$: spatially, between their median concentrations, and temporally, according to their seasonal cycles. In addition, the quality of annual maximum $NO_3$ concentration was in-phase with maximum flow when the base flow index was low, but this synchrony disappeared when flow flashiness was lower. These DOC-$NO_3$ seasonal cycle types were related to the mixing of flowpaths combined with the spatial variability of their respective sources and to local biogeochemical processes. The annual maximum SRP concentration occurred during the low-flow period in nearly all catchments. This likely resulted from the dominance of P point sources. The approach shows that despite the relatively low frequency of public water quality data, such databases can provide consistent pictures of the spatio-temporal variability of water quality and of its drivers as soon as they contain a large number of catchments to compare and a sufficient length of concentration time series.

**1 Introduction**

As a condition for human health, food production, and ecosystem functions, water quality is recognized as "one of the main challenges of the 21st century" (FAO and WWC, 2015; UNESCO, 2015), and potential impacts of climate change on water quality are even more challenging (Whitehead et al., 2009). To better estimate and reduce human impact on water quality, water scientists are expected to provide integrated understanding of multiple pollutants (Cosgrove and Loucks, 2015). Eutrophication risks (Dodds and Smith, 2016) are considered the main factors that decrease the quality of surface water, according to objectives set by the European Union Water Framework Directive. Mitigating the problem of eutrophication involves considering at least the three major elements: carbon (C), nitrogen (N), and phosphorus (P) (Le Moal et al., 2019).

In addition, the quality of headwater catchments have been studied less than large rivers (Bishop et al., 2008), despite their influence on downstream water quality (Alexander et al., 2007; Barnes and Raymond, 2010; Bol et al., 2018) and higher spatial variability in their concentrations (Abbott et al., 2018a; Temnerud and Bishop, 2005). One reason for this is that most water quality monitoring networks coincide with the location of drinking-water production facilities, which explains why they focus on large rivers. Nonetheless, investigating spatial variability in upstream water quality is relevant for understanding what causes it to degrade, targeting locations with the greatest disturbances, and identifying which remediation policies would be most cost effective.

In non-agricultural headwater catchments, spatial variability in dissolved organic C (DOC) concentrations in streams has been related to topography, wetland coverage, and soil properties such as clay content or pH (Andersson and Nyberg, 2008; Brooks et al., 1999; Creed et al., 2008; Hytteborn et al., 2015; Musolff et al., 2018; Temnerud and Bishop, 2005; Zarnetske et al., 2018). Stream DOC concentrations and composition in agricultural and urbanized areas also generally differ greatly from those in semi-natural or pristine catchments (Graeber et al., 2012; Gücker et al., 2016). Over large gradients of human impact (e.g. from undisturbed to urban catchments), the cover of agricultural and urban land uses often appears as a key factor that explains differences in stream chemistry of C, N, and P species (e.g. Barnes and Raymond, 2010; Edwards et al., 2000; Mutema et al., 2015) and even silica (Onderka et al., 2012). Conversely, in mostly undisturbed catchments (Mengistu et al., 2014) or ~~mostly in~~ rural catchments where human pressure are low (Heppell et al., 2017; Lintern et al., 2018)~~.~~ – "natural" controls such as topography, geology, and flow paths are more frequently highlighted as the main factors that explain spatial variability in C, N and P.

Besides being spatially variable, C, N, and P concentrations also vary temporally. The variability of concentrations with flow has been described in several studies using concentration-flow relationships at event (Fasching et al., 2019) or inter-annual to long-term scales (Basu et al., 2010; 2011; Moatar et al., 2017). Concentrations also vary seasonally in streams and rivers (Aubert et al., 2013; Dawson et al., 2008; Duncan et al., 2015; Exner-Kittridge et al., 2016; Lambert et al., 2013), as does the composition of dissolved organic matter (Griffiths et al., 2011; Gücker et al., 2016). This seasonality can also be spatially structured. Several studies showed that the relative importance of catchment characteristics on water concentrations or loads varied by season because nutrient sources and biological and physico-chemical processes that influence nutrient mobilization

and transfer in catchments (e.g. vegetation uptake, in-stream biomass production, denitrification) changed with the hydrological, light and temperature conditions (Ågren et al., 2007; Fasching et al., 2016; Gardner and McGlynn, 2009). Some variability in seasonal patterns of dissolved C, N, and/or P concentrations among headwater catchments has been reported (e.g. Van Meter et al., 2019; Abbott et al., 2018b; Duncan et al., 2015; Martin et al., 2004). Identifying these patterns is relevant

from a management viewpoint as they may indicate changes in the locations of C, N, or P sources or their transfer pathways.

Thus, to date, analysis of spatial variability in water quality at the headwater scale:

1) is usually restricted to one element, although multi-element approaches are becoming more frequent (Edwards et al., 2000; Heppell et al., 2017; Lintern et al., 2018; Mengistu et al., 2014; Mutema et al., 2015),

2) is particularly rare for headwater catchments with similar human pressures (e.g. intensive farming), despite the high variability in water quality sometimes observed among them (e.g. Thomas et al., 2014),

3) often uses mean annual values (concentration or load) to describe spatial variability in water quality among catchments, with little or no analysis of seasonal patterns despite their frequent occurrence (Van Meter et al., 2019; Abbott et al., 2018b; Liu et al., 2014; Halliday et al., 2012; Mullholland et al. 1997), and

4) is usually restricted to a few catchments: multiple-catchment studies on multiple elements are uncommon, despite their ability to identify dominant controlling factors better.

We studied the spatial variability and seasonal variation in water quality of 185 headwater catchments (from 5-179 km²) draining Brittany, an intensively farmed region of France. Our analysis focuses on dissolved C, N, and P concentrations as DOC, nitrate ($NO_3$), and soluble reactive P (SRP), respectively. We hypothesized that:

1) Human (i.e. rural and urban) pressures determine spatial variability in $NO_3$ and SRP concentrations (Preston et al., 2011; Melland et al., 2012; Dupas et al., 2015a; Kaushal et al., 2018), while soil and climate characteristics, including light and temperature along the stream, determine that in DOC and possibly SRP (Lambert et al., 2013; Humbert et al., 2015; Gu et al., 2017).

2) Seasonal variations in water quality provide information about spatial variability in biogeochemical sources and/or

reactivity in catchments as a function of changes in water pathways and are correlated in part with spatial variability in concentrations and loads.

We selected headwater catchments for which relevant time series of DOC, $NO_3$, and SRP concentrations and stream flow were available (10 years of consecutive data measured at least monthly). In addition to estimating interannual loads, we calculated

concentration metrics for each element to assess the spatial variability and temporal variation in water quality. Generalized Additive Models (GAMs) were applied to the time series to highlight average patterns of seasonal variation. Correlations between the water quality metrics and the geological, soil, climatic, hydrological, land cover, and human pressure characteristics of the corresponding headwater catchments were evaluated using rank correlation analyses.

## 2 Materials and Methods

### 2.1 Study area

Brittany is a 27,208 km² region in western France. Its bedrock is composed mainly of a crystalline substratum dominated by granite and schist (Supplement S1b). Its topography is moderate, with elevation ranging from 0-330 m a.s.l. Its climate is temperate oceanic, with precipitation ranging from 531 mm.yr$^{-1}$ in the east to 1070 mm.yr$^{-1}$ on the western coasts (regional median of 723.0 mm.yr$^{-1}$) (Supplement S1a), and a mean annual temperature of 12°C. The regional hydrographic network is dense, with a mean density of 1 km.km$^{-2}$. ~~Its intensive agriculture has a strong influence on land use and agri-food production.~~ Overall, 56.6% of the region was Utilized Agricultural Area (UAA) in 2017 (data from DREAL Bretagne, Brittany's Agency for Environment, Infrastructure, and Housing), which represented 6% of national UAA in 2016. Of total French production, Brittany produces 17.4% of milk and dairy products, 20% of pork products, and 17% of eggs and poultry (Brittany Chamber of Agriculture, 2016 data). At the canton (administrative district) scale, mean N and P surpluses are high and have high spatial variability (standard deviation (SD)): 50.01 ± 26.59 kg N.ha$^{-1}$.yr$^{-1}$ and 22.52 ± 12.66 kg P.ha$^{-1}$.yr$^{-1}$ (Supplement S1e,f). The region has a population of ca 3.3 million inhabitants (data 2017), some scattered throughout the region, and some concentrated in a few cities and near the coasts (Supplement S1c,d).

### 2.2 Stream data selection and headwater characteristics

Water quality data consisted of time series of DOC, NO$_3$, and SRP concentrations, extracted from two public monitoring networks – OSUR (Loire-Brittany Water Agency, 554 sites) and HYDRE/BEA (DREAL Bretagne, ca. 1964 sites), measured for regulatory monitoring, regional contracts, or specific programs. Concentrations were measured from grab samples. Headwater catchments were selected according to the following two criteria: (i) independence, with no overlap of the drained areas of the water-quality stations selected, and (ii) availability of at least 80 measurements of DOC, NO$_3$, and SRP concentrations at the same station (after removing outliers based on expert knowledge, i.e. values > 200 mg N.L$^{-1}$ or 5 g P.L$^{-1}$) over 10 calendar years (2007-2016). We selected 185 stations (83% and 17% from OSUR and HYDRE/BEA, respectively) (hereafter, "concentration (C) stations"), which had mean frequencies of 12, 14, and 11 analyses per year for DOC, NO$_3$, and SRP, respectively. We checked that there was no bias in the timing of concentration data: OSUR database has fixed and regular sampling frequencies while we noticed a few time series where summer periods were less sampled in the HYDRE/BEA data for some years only.

Each C station was paired with a hydrometric station (Q). Observed daily streamflow data from the national hydrometric network (http://hydro.eaufrance.fr/) were used when draining headwater catchments for C and Q stations shared at least 80% of their areas (25% of cases). When observed Q data were not available, or at a frequency less than 320 measurements per year from 2007-2016 (75% of cases), discharge data were simulated using the GR4J model (Perrin et al., 2003). The headwater catchments selected and their associated C and Q stations were distributed throughout Brittany (Fig. 1).

The 185 headwater catchments selected cover ca. 32% of Brittany's area. Despite having a similar hydrographic context dominated by subsurface flow, the catchments have large differences in topography, geology, hydrology, and diffuse and point-source pressures of N and P. We used a set of catchment descriptors to quantify this variability (Table−1) (see Supplemental S2 for their statistical distribution and S3 for their correlations). The descriptors selected included a set of spatial metrics for element sources (e.g. land use, pressure, soil contents) and for mobilization and retention processes (e.g. hydrology, climate, topography, geology, and soil properties).

The headwater catchments range in area from 5-179 km² (median of 38 km²), and the density of each one's hydrographic network ranges from 0.47-1.49 km.km⁻² (median of 0.90 km.km⁻²). Strahler stream order is 3 for 36% of the catchments, 2 for 18%, 4 for 17%, and 1 for 11%. Substrate composition is dominated by schists/micaschists (44%) or granites/gneisses (31%). In the topsoil horizon (0-30 cm), the soil organic C content varies greatly from 18.6-565.4 $g.kg^{-1}$ (median of 126.9 $g.kg^{-1}$), while the total P (Dyer method) content varies from 0.6-1.4 $g.kg^{-1}$ (median of 0.9 $g.kg^{-1}$). Land use is largely agricultural, although some catchments have high percentages of forested and urbanized areas. Riparian wetlands cover 12.3-36.3% of catchment area (median of 22.4%), forest covers 1.3-55.7% (median of 13.2%), pasture covers 10.3-46.7% (median of 25.6%), summer crops cover 6.5-50.3% (median of 27.8%), and winter crops cover 7.0-51.0% (median of 22.7%). The N and P surplus (potential diffuse agricultural sources) vary from 12.9-96.0 kg $N.ha^{-1}.yr^{-1}$ (median of 47.7) and 2.8-63.2 kg $P.ha^{-1}.yr^{-1}$ (median of 18.9), respectively. Urban areas cover 1.3-31.8% of the headwater catchments (median of 6%), with point-source input estimates ranging from 0-6.2 kg N. $ha^{-1}.yr^{-1}$ and 0-0.626 kg P. $ha^{-1}.yr^{-1}$. These data illustrate relative diversity in human pressures among the catchments despite a regional context of intensive agriculture. The daily mean flow (Qmean) varies from 4.8-24.5 $l.s^{-1}.km^{-2}$ (median of 10.8 $l.s^{-1}.km^{-2}$), the median of annual minimum of monthly flows (QMNA) varies from 0.2-5.9 $l.s^{-1}.km^{-2}$, and the flow flashiness index (W2), defined as the percentage of total discharge that occurs during the highest 2% of flows (Moatar et al., 2020), ranges from 10-28%.

### 2.3 Data analysis

### 2.3.1 Concentration and load metrics

To analyze spatial variability in DOC, $NO_3$, and SRP concentrations in streams, we calculated their 10[th], 50[th], and 90[th] percentiles of concentration (C10, C50, and C90, respectively) for each headwater catchment from 2007-2016. We also calculated the ratio of the coefficient of variation (CV) of mean concentration ($CVc_{mean}$) and to that of mean flow ($CVq_{mean}$) to compare spatial variabilities in concentrations and stream flow. We estimated interannual loads for a 10-year period (2007-2016), with 8-12 C-Q values per year. However, a 5-year period (2010-2014) was considered to analyze the spatial variability because it minimized data gaps (in C and Q time series) among all stations simultaneously.

To calculate interannual DOC, $NO_3$, and SRP loads for each headwater catchment, we tested different methods and selected the most suitable, depending on the reactivity of the element with flow. When C-Q relationships were relatively flat or diluted

(NO$_3$) or slowly mobilized (DOC) during high flow (Q>Q50) , we used the discharge weighted concentration (DWC) method (Eq. 1), which estimates loads with lower uncertainties (Moatar and Meybeck, 2007; Raymond et al., 2013):

$$DWC = \frac{k}{A} \times \frac{\sum_{i=1}^{n} C_i Q_i}{\sum_{i=1}^{n} Q_i} \overline{Q} \tag{1}$$

where DWC is the mean of annual loads (kg.y$^{-1}$.ha$^{-1}$), $C_i$ is the instantaneous concentration (mg.l$^{-1}$), Q$_i$ is the corresponding flow rate (m$^3$.s$^{-1}$), $\overline{Q}$ is the mean annual flow rate calculated from daily data (m$^3$.s$^{-1}$), A is the area of the headwater catchment (m²ha), k is a conversion factor (31557.631536), and n is the number of C-Q pairs per year.

The loads estimated by the DWC method were corrected for bias (Moatar et al., 2013). Precisions were calculated from the number of samples (n), number of years, export regime exponent (b$_{50high}$), and W2 (Moatar et al., 2020).

To calculate SRP loads, regression methods were more suitable (because of strong concentration patterns when stream flow increases). We averaged the loads estimated by two regression methods developed by Raymond et al. (2013) – Integral Regression Curve (IRC) and Segmented Regression Curve (SRC) – both based on a regression between concentration and
175 flow:

$$IRC = \frac{k'}{A} \times \sum_{i=1}^{n} C_i Q_i \tag{2}$$

$$SRC = \frac{k'}{A} \times \left( \sum_{i=1}^{m1} C_{infi} Q_i + \sum_{i=1}^{m2} C_{supi} Q_i \right) \tag{3}$$

where IRC and SRC are the mean of annual loads (kg.y$^{-1}$.ha$^{-1}$); C$_i$, C$_{supi}$, and C$_{infi}$ are instantaneous concentrations estimated
by the regression curves (mg.l$^{-1}$); C$_{supi}$ and C$_{infi}$ are concentrations estimated for of flows above and below the median flow, respectively; n = 365 days; m1 and m2 are numbers of days with daily flows below and above the median flow, respectively; and $k'$ is a conversion factor (86.4); and A is the area of the headwater catchment (ha).

### 2.3.2 Seasonal signal

Seasonal dynamics of discharge and solute concentrations were modeled using GAMs (Wood, 2017), which can estimate smoothed seasonal dynamics from time series (Musolff et al., 2017). The smoothing function was a cyclic cubic spline fitted to the month of the year (1-12); thus, the ends of the spline were forced to be equal, using the R package mgcv. We did not consider a long-term trend in the time series over the 10 years, for two reasons. First, significant long-term trends (according to Man-Kendall tests) had low slopes: mean Theil-Sen slopes ranged from -3% to 0% of the median concentration (while mean
seasonal relative amplitudes exceeded 50%). Second, performance of the GAMs did not increase significantly when a long-term trend was added: the mean adjusted coefficient of determination (Rsq) increased from 0.16 to 0.18 for DOC and from 0.30 to 0.40 for NO$_3$. We considered a seasonal dynamic to exist when the GAM adjusted coefficient of determination was

greater than 0.10. ~~This method provides SI metrics on seasonality, which can be easily be linked to different geographical variables. However, it cannot take into account catchments without seasonality.~~

Seasonal dynamics of the concentrations of the three solutes (DOC, $NO_3$, and SRP) and river discharge were then analyzed using five metrics calculated from the daily simulations of the GAMs. The first three were the annual amplitude (Ampli; i.e. annual maximum minus annual minimum), and the mean time in which annual maximum and minimum concentrations occurred (MaxPhase and MinPhase, respectively; in months from 1 January). The next was Ampli standardized by the corresponding mean concentration to compare the three solutes. The last metric was a seasonality index (SI), which measures the relative importance of summer (1 June to 31 July) concentrations compared to winter (15 January to 15 March) concentrations of an element, as follows (Eq. 4):

$$SI = \frac{C_{winter} - C_{summer}}{C_{winter} + C_{summer}} \tag{4}$$

where $C_{winter}$ and $C_{summer}$ are the averages of winter and summer concentrations, calculated from daily values from fitted GAM. Positive values of SI (near 1) indicate that $C_{winter} > C_{summer}$, while negative values (near -1) indicate that $C_{winter} < C_{summer}$. We considered that SI values close to 0 (from -0.1 to 0.1) indicated that $C_{winter}$ equaled $C_{summer}$. The SI integrates both amplitude and phasing features of the seasonal signal. These five metrics, obtained from daily simulations of the GAMs, are linked to geographical variables (2.3.2), even if particular solute in some catchments do not present any seasonality.

**2.3.2 Statistical analyses**

To compare the concentration metrics of the elements, a multivariate analytical approach, principal component analysis (PCA), was performed for the 9 variables of concentration percentiles (C10, C50, and C90) of DOC, $NO_3$, and SRP for the dataset of 185 headwater catchments. PCA was chosen despite its assumption of~~es~~ linear relationships between variables, because it provides a graphical representation of correlations between variables or groups of variables and their contributions to the variance. To identify dominant drivers of spatial variability in concentration percentiles, seasonality, and loads of DOC, $NO_3$, and SRP, we calculated Spearman's rank correlation ($r_s$) between these water-quality metrics and the descriptors of the headwater catchments (Table 1). We considered a rank correlation to be significant if the corresponding p-value was $\leq 0.05$. All analyses were performed using R software (v. 3.6.1) with packages mgcv, hydroGOF, hydrostats, FactoMineR, tidyverse, lubridate, reshape2, plyr, ggcorrplot, and ggplot2 (Grolemund and Wickham, 2011; Le et al., 2008; Wickham, 2016, 2011; Wood, 2017; Zambrano-Bigiarini, 2020).

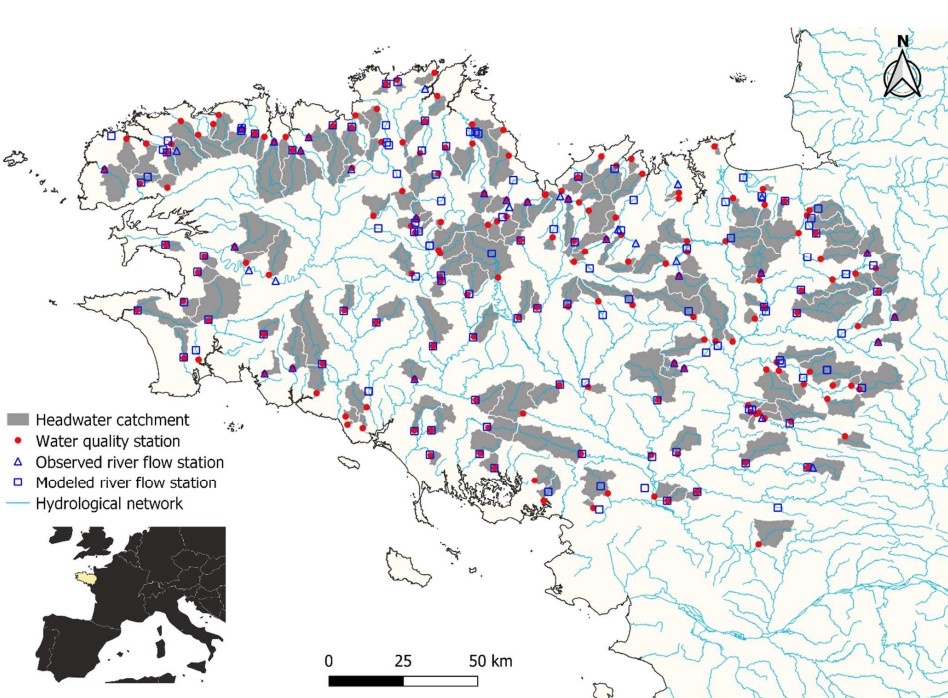

**Figure 1. Locations of the 185 study headwater catchments where dissolved organic carbon, nitrate, and soluble reactive phosphorus concentrations were monitored monthly at the outlet from 2007-2016, and paired discharge stations where daily records of stream flow were available from observations or modeling.**

Table 1. Headwater catchment descriptors identified as potential explanatory variables of spatial variability and temporal variation in dissolved organic carbon (DOC), nitrate (NO₃), and soluble reactive phosphorus (SRP) in stream and river water. $Topo\_i = log \frac{\alpha}{tan\beta}$ , (Beven and Kirkby, 1979), where $\alpha$ is the drainage area (ha) and $\beta$ is the downstream slope (%), (Merot et al., 2003).

[a] there are 3 classes of soil thickness: 40-60 cm, 60-80 cm, 80-100 cm and >100 cm. [b] Winter crops have a winter plant cover and a phenological maximum in April (wheat, barley, rapeseed). [c] Summer crops correspond to bare winter soils and a phenological maximum in early summer (corn).

| Type | Descriptor name | Unit | Definition | Source |
|------|------|------|------|------|
| Topography | Area | km² | Drainage area of the monitoring station | Web Processing Service "Service de Traitement de Modèles Numériques de Terrain" and DEM 50 m by IGN |
| | Elevation | m | Mean elevation of headwater catchment | DEM 25 m by IGN |
| | Density_hn | km.km⁻² | Density of the hydrographic network | |
| | Topo_i | log(m²) | Downstream topographic index of the headwater | BD Carthage by IGN |
| | IDPR | - | Hydrographic Network Development and Persistence Index | http://infoterre.brgm.fr/ BRGM data and geoservices portal (Mardhel and Gravier, 2004) |
| Geology | Granite_pm | % | Percentage of granite and gneiss area | Web Mapping Service "Carte des Sols de Bretagne" by UMR 1069 SAS INRAE Agrocampus Ouest http://www.sols_de_bretagne.fr/ |
| | Schist_pm | % | Percentage of schist and micaschist area | |
| | Other_pm | % | Percentage of various geological substrata | |
| Soil | Erosion | % | Percentage of area with high to very high erosion risk (derived from land use, topography and soil properties) | Erosion risk map estimated from MESALES by GIS Sol, INRAE from Colmar et al. (2010) |
| | OC_soil | g.kg⁻¹ | Organic carbon content in the topsoil horizon (0-30 cm) | Web Mapping Service from BDAT database, Saby et al. (2015) by GIS Sol |
| | Thick_soil | cm | Classes of dominant soil thickness[a] | Web Mapping Service "Carte des Sols de Bretagne" by UMR 1069 SAS INRAE Agrocampus Ouest |
| | TP_soil | g.kg⁻¹ | Total phosphorus content in the topsoil horizon (0-30 cm) | Web Mapping Service from BDAT database by GIS Sol |
| Land use | SummerCrop | % | Percentage of summer crop[b] land | |
| | WinterCrop | % | Percentage of winter crop[c] land | OSO database, CESBIO, land cover map 2016 (1 ha) from http://osr-cesbio.ups-tlse.fr/~oso/ |
| | Forest | % | Percentage of forest land | |
| | Pasture | % | Percentage of pasture land | |
| | Urban | % | Percentage of urban land | |
| | Wetland | % | Percentage of potential wetlands | Web Mapping Service "Enveloppe des milieux potentiellement humides de France réalisée par les laboratoires Infosol et UMR SAS" by UMR 1069 |

| Type | Descriptor name | Unit | Definition | Source |
|---|---|---|---|---|
| | | | | SAS INRAE - Agrocampus Ouest / US 1106 InfoSol INRAE |
| Diffuse and point N and P sources | N_surplus | kg.ha⁻¹.yr⁻¹ | Nitrogen surplus (= the maximum quantity on a given agricultural area that is likely to be transferred to the stream network) | CASSIS-N estimates by (Poisvert et al., 2017) from |
| | P_surplus | kg.ha⁻¹.yr⁻¹ | Phosphorous surplus | NOPOLU estimates by (SoeS, 2013) |
| | N_point | kg.ha⁻¹.yr⁻¹ | Sum of nitrogen loads from domestic and industrial point sources | Data from Loire-Bretagne Water Agency data (2008-2012) |
| | P_point | kg.ha⁻¹.yr⁻¹ | Sum of phosphorus loads from domestic and industrial point sources | Data from Loire-Bretagne Water Agency (2008-2012) |
| Hydrology | Qmean | l.s⁻¹.km⁻² | Interannual mean flow | |
| | QMNA | l.s⁻¹.km⁻² | Median of annual minimum monthly specific discharge | Calculated from flow data observations: HYDRO regional database by DREAL Bretagne & GR4J simulations (Perrin et al., 2003) |
| | BFI | % | Base flow index (Lyne et Hollick, 1979) | |
| | W2 | % | Percentage of total discharge that occurs during the highest 2% of flows (Moatar et al., 2013) | |
| | Rainfall | mm.yr⁻¹ | Mean effective rainfall from 2008-2012 | SAFRAN database (8 km²) by Météo France |

**Table 1. Headwater catchment descriptors identified as potential explanatory variables of spatial variability and temporal variation in dissolved organic carbon (DOC), nitrate (NO₃), and soluble reactive phosphorus (SRP) in stream and river water. meanTWI $= log \frac{\alpha}{tan \beta}$, (Beven and Kirkby, 1979), where $\alpha$ is the drainage area (ha) and $\beta$ is the downstream slope (%), (Merot et al., 2003). [a] there are 3 classes of soil thickness: 40-60 cm, 60-80 cm, 80-100 cm and >100 cm. [b] Winter crops have a winter plant cover and a phenological maximum in April (wheat, barley, rapeseed). [c] Summer crops correspond to bare winter soils and a phenological maximum in early summer (corn).**

| Type | Descriptor name | Unit | Definition | Source |
|---|---|---|---|---|
| Topography | Area | km² | Drainage area of the monitoring station | Web Processing Service "Service de Traitement de Modèles Numériques de Terrain" and DEM 50 m by IGN |
| | Elevation | m | Mean elevation of headwater catchment | DEM 25 m by IGN |
| | Density_hn | km.km⁻² | Density of the hydrographic network | BD Carthage by IGN |
| | meanTWI | cf. legend log(Ha) | Average topographic wetness index of the headwater catchment | DEM 25 m by IGN |
| | IDPR | - | Hydrographic Network Development and Persistence Index | http://infoterre.brgm.fr/ BRGM data and geoservices portal (Mardhel and Gravier, 2004) |
| Geology | Granite_pm | % | Percentage of granite and gneiss area | Web Mapping Service "Carte des Sols de Bretagne" by UMR 1069 SAS INRAE - Agrocampus Ouest http://www.sols-de-bretagne.fr/ |
| | Schist_pm | % | Percentage of schist and micaschist area | |
| | Other_pm | % | Percentage of various geological substrata | |
| Soil | Erosion | % | Percentage of area with high to very high erosion risk (derived from land use, topography and soil properties) | Erosion risk map estimated from MESALES by GIS Sol, INRAE from Colmar et al. (2010) |
| | OC_soil | g.kg⁻¹ | Organic carbon content in the topsoil horizon (0-30 cm) | Web Mapping Service from BDAT database, Saby et al. (2015) by GIS Sol |

| | Variable | Units | Description | Source |
|---|---|---|---|---|
| | Thick_soil | cm | Classes of dominant soil thickness[a] | Web Mapping Service "Carte des Sols de Bretagne" by UMR 1069 SAS INRAE - Agrocampus Ouest |
| | TP_soil | g.kg$^{-1}$ | Total phosphorus content in the topsoil horizon (0-30 cm) | Web Mapping Service from BDAT database by GIS Sol |
| Land use | SummerCrop | % | Percentage of summer crop[b] land | |
| | WinterCrop | % | Percentage of winter crop[c] land | OSO database, CESBIO, land-cover map 2016 |
| | Forest | % | Percentage of forest land | (1 ha) from http://osr-cesbio.ups-tlse.fr/~oso/ |
| | Pasture | % | Percentage of pasture land | |
| | Urban | % | Percentage of urban land | |
| | Wetland | % | Percentage of potential wetlands | Web Mapping Service "Enveloppe des milieux potentiellement humides de France réalisée par les laboratoires Infosol et UMR SAS" by UMR 1069 SAS INRAE - Agrocampus Ouest / US 1106 InfoSol INRAE |
| Diffuse and point N and P sources | N_surplus | kg.ha$^{-1}$.yr$^{-1}$ | Nitrogen surplus (= the maximum quantity on a given agricultural area that is likely to be transferred to the stream network) | CASSIS-N estimates by (Poisvert et al., 2017) from https://geosciences.univ-tours.fr/cassis/login |
| | P_surplus | kg.ha$^{-1}$.yr$^{-1}$ | Phosphorous surplus | NOPOLU estimates by (SoeS, 2013) |
| | N_point | kg.ha$^{-1}$.yr$^{-1}$ | Sum of nitrogen loads from domestic and industrial point sources | Data from Loire-Bretagne Water Agency data (2008-2012) |
| | P_point | kg.ha$^{-1}$.yr$^{-1}$ | Sum of phosphorus loads from domestic and industrial point sources | Data from Loire-Bretagne Water Agency (2008-2012) |
| Hydrology | Qmean | l.s$^{-1}$.km$^{-2}$ | Interannual mean flow | |
| | QMNA | l.s$^{-1}$.km$^{-2}$ | Median of annual minimum monthly specific discharge | Calculated from flow data observations: HYDRO regional database by DREAL Bretagne & GR4J simulations (Perrin et al., 2003) |
| | BFI | % | Base flow index (Lyne et Hollick, 1979) | |
| | W2 | % | Percentage of total discharge that occurs during the highest 2% of flows (Moatar et al., 2013) | |
| | Rainfall | mm.yr$^{-1}$ | Mean effective rainfall from 2008-2012 | SAFRAN database (8 km²) by Météo France |

## 3 Results

### 3.1 Spatial variability in concentrations and loads

The C50 of the 185 headwater catchments ranged from 2-14.6 mg C.l$^{-1}$ for DOC, 0.9-15.8 mg N.l$^{-1}$ for NO$_3$, and 8-241 µg P.l$^{-1}$ for SRP (with 75% of the SRP C50 < 64 µg P.l$^{-1}$). The C50 displayed spatial gradients: rivers with DOC concentrations > 5 mg C.l$^{-1}$ were located in eastern Brittany, while the highest NO$_3$ concentrations were located on the west coast (Fig. 2). In contrast, the highest concentrations of SRP (C50 > 68 µg P.l$^{-1}$) were located in northern Brittany.

The two first axes of the PCA (Fig. 23aSupplemental S4a) performed on the percentiles of DOC, NO$_3$, and SRP concentrations of the 185 headwater catchments explained 58% of the variance and revealed three important points. First, percentiles (C10, C50, or C90) were grouped by solute, showing that the spatial organization remained the same regardless of the concentration percentile (Spearman rank correlations between the three indices always greater than 0.56 for all elements). Second, there was a negative correlation between the C50 of DOC and NO$_3$ concentrations ($r_s$ = -0.58; Fig. 3b and Supplemental S3, S82Supplemental S4b). Third, SRP concentrations had an orthogonal relation compared to DOC and NO$_3$ concentrations ($r_s$ close to zero).

The ratios of mean concentration ($CVc_{mean}$) to mean flow ($CVq_{mean}$) were < 1 for DOC and $NO_3$ (Table 2), indicating that concentrations varied less in space than in flow, and vice-versa for SRP.

For DOC and $NO_3$, Ampli was not correlated significantly with C50, but it was with C90 (Fig. 4 and Supplemental S83). For SRP, correlations between Ampli and the percentiles were high, with $r_s$ > 0.85 for C50 and C90 (Fig. 34, Supplemental S8). The SI and phases, calculated on the catchments for which a GAM can be fitted, i.e. presenting a seasonal feature, were correlated more with C10 for DOC (n=107) and $NO_3$ (n=98) (negatively for SI and positively for the phases), and more with C90 for SRP (n=118) (negatively, for SI only).

Mean (± 1 SD) interannual loads had high spatial variabilities − 20.71 ± 10.52 kg C.$ha^{-1}$.$yr^{-1}$ for DOC, 27.48 ± 18.51 kg N.$ha^{-1}$.$yr^{-1}$ for $NO_3$, and 0.315 ± 0.11 kg P.$ha^{-1}$.$yr^{-1}$ for SRP − which differed from those observed for concentrations (Fig. 2). Unsurprisingly, interannual loads of the three solutes were significantly (p<0.001) and strongly correlated with annual water fluxes (Pearson r = 0.88 for DOC, 0.90 for $NO_3$, and 0.75 for SRP). There were weak but significant positive correlations between mean interannual loads and seasonality indices (Ampli, SI) or C90 for DOC (Fig. 34). Mean interannual loads of $NO_3$ were significantly and positively correlated with C10 and C50, and negatively with its seasonality indices. The strongest significant correlation was found between mean interannual loads and concentration percentiles for SRP.

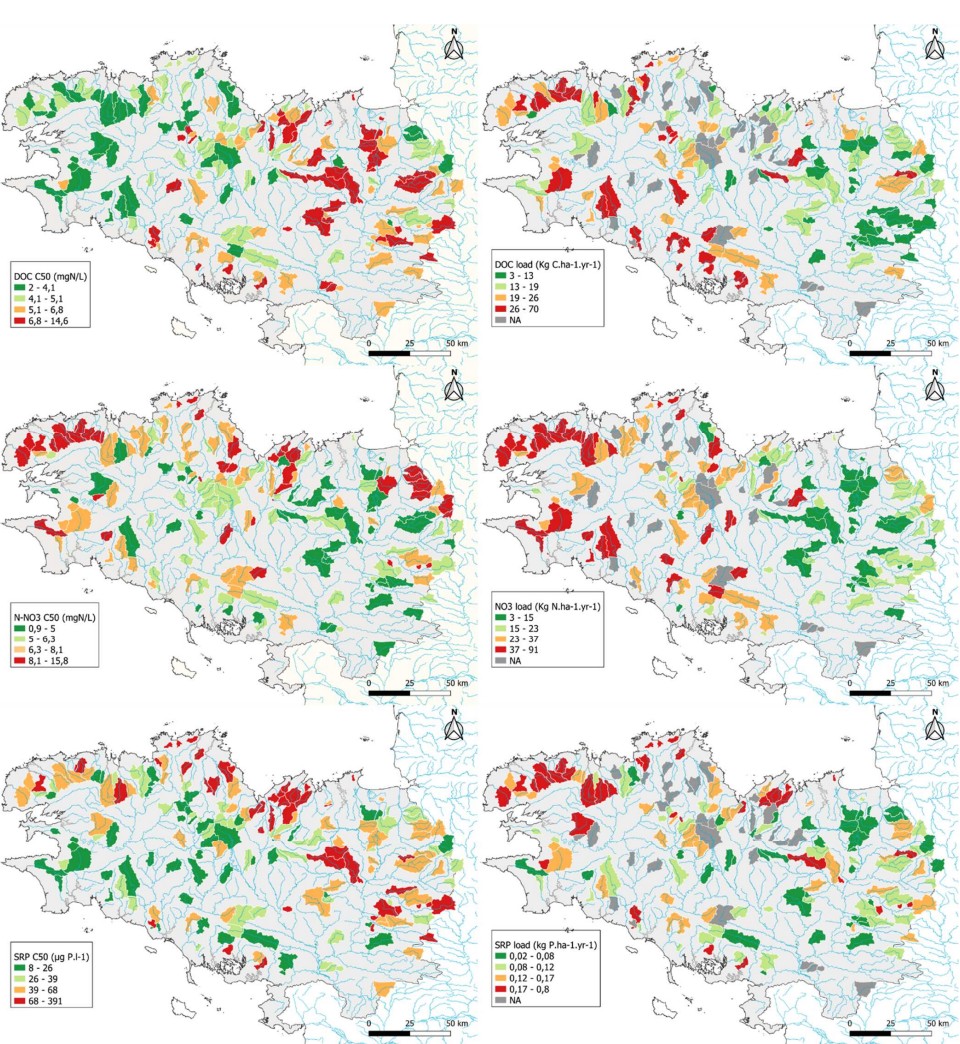

**Figure 2. Map of median (left) concentrations C50 and (right) loads of dissolved organic carbon (DOC), nitrate N (N-NO₃), and soluble reactive phosphorus (SRP) for the 185 streams. The catchments in gray did not meet the criteria to estimate a mean average interannual load. Classes in the legends have equal numbers of catchments.**

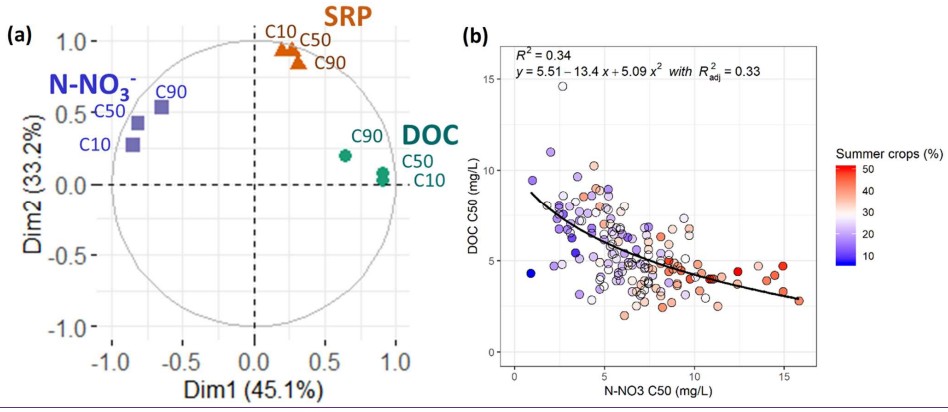

Figure 3. (a) Principal component analysis of $10^{th}$, $50^{th}$, and $90^{th}$ percentiles (C10, C50 and C90) of nitrate (N-NO₃), dissolved organic carbon (DOC), and soluble reactive phosphorus (SRP) concentrations for the 185 headwater catchments analyzed; (b) Correlation between the medians (C50) of DOC and N-NO₃ concentrations for the 159 catchments in which DOC and NO₃ were monitored from 2007-2017. The color gradient indicates the percentage of catchment area covered by summer crops.

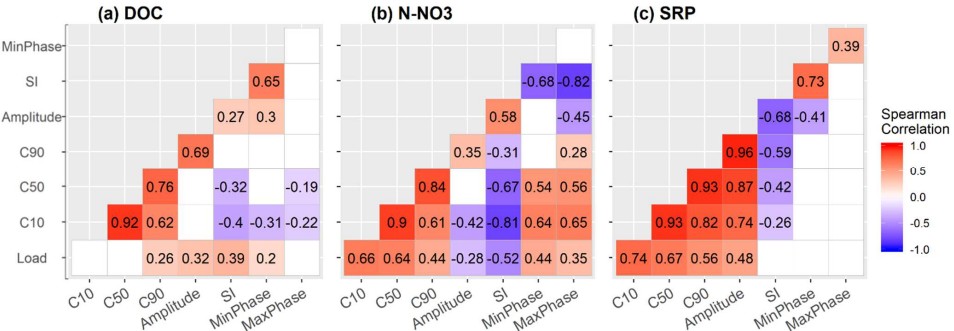

Figure 43. Matrices of Spearman's rank correlations of water quality (load, concentration percentiles ($10^{th}$ (C10), $50^{th}$ (C50), and $90^{th}$ (C90)), and seasonality metrics) for (a) dissolved organic carbon (DOC), (b) nitrate N (N-NO₃), and (c) soluble reactive phosphorus (SRP) (c). Only significant ($p \leq 0.05$) values are shown.

**Table 2. Coefficients of variation (spatial variability among catchments) of flow-weighted mean concentration (CVcmean) and mean stream flow (CVqmean), and the value of their ratio, for dissolved organic carbon (DOC), nitrate ($NO_3$), and soluble reactive phosphorus (SRP).**

| Parameter | CVcmean | CVqmean | CVcmean:CVqmean |
|-----------|---------|---------|-----------------|
| DOC | 0.2954 | 0.4614 | 0.6403 |
| $NO_3$ | 0.3285 | 0.4709 | 0.6976 |
| SRP | 0.9207 | 0.4743 | 1.9412 |

### 3.2 Characterization of concentrations seasonality

#### 3.2.1 Performance of GAMS

Of the 185 catchments, GAMs were fitted for 159 to DOC concentrations time series, 168 to $NO_3$ concentrations time series, 162 to SRP concentrations time series, and 185 to discharge time series. The cases for which fitting was not possible

corresponded to those with no seasonal cyclicity or with excessive interannual variability. The percentage of variance explained by the GAM varied by site and solute. Fitting performed best for $NO_3$, followed by SRP and then DOC: the means and SDs of the adjusted Rsq were $0.30 \pm 0.18$, $0.16 \pm 0.11$, and $0.22 \pm 0.15$ for $NO_3$, DOC, and SRP, respectively (Supplemental S45 and S65), and the percentages of catchment for which the fitted model had Rsq > 0.20 were 67%, 52% and 38%, respectively. Metrics calculated from monthly data differed only moderately from those calculated from sub-monthly data

(Supplemental S76), which tended to validate the approach of using monthly data.

#### 3.2.2 Types of seasonal cyclicity in DOC, $NO_3$, and SRP

Most of the catchments had a seasonal concentration cycle: 85%, 71%, 78%, for $NO_3$, DOC, SRP concentration respectively and 100% of them had a seasonal discharge cycle (Fig. 54). Means and SDs of the standardized Ampli were $0.59 \pm 0.46$ for

$NO_3$, $0.53 \pm 0.30$ for DOC, $0.79 \pm 0.14$ for SRP, and $1.99 \pm 0.38$ for discharge. The distribution of the calculated seasonality indices is provided in Supplemental S78.

The annual phases for discharge were more stable among all catchments than those for concentrations. The highest discharge period was centered on mid-February (winter) and the lowest discharge period on September. A strong gradient of hydrological dynamics was observed among catchments (Fig. 54d and Supplemental S78). The highest W2 was associated with both severe

low-flow discharge and many high discharge events. Values of $Q_{mean}$, BFI, W2, and QMNA clearly followed an east-west gradient (not shown). Because of similar seasonal discharge dynamics in all catchments, SI can be used to describe the seasonal dynamics of a concentration relative to those of discharge. When SI was positive, the concentration seasonality was in-phase with discharge; when negative, the concentration seasonality was out-of-phase with discharge (Fig. 54).

Most of the catchments had opposite dynamics for DOC and $NO_3$. For 90% of them, Pearson correlation between the daily GAM estimates of DOC and $NO_3$ was negative, and for 50% of the catchments, less than -0.79. The remaining 10% of catchments (15) had low Ampli of DOC and $NO_3$. The DOC and $NO_3$ concentrations had out-of-phase seasonal cycles, as shown by the negative correlation between SI and DOC or $NO_3$ for all catchments that had a significant seasonality in these concentrations (Fig. 65; $R^2$ = 0.62). We classified two types of catchments according to their seasonality in both DOC (MinPhase) and $NO_3$ (MaxPhase) concentrations and consistent with the SI (Fig. 65, Supplemental S78). $NO_3$ MaxPhase and DOC MinPhase that occurred before 1 May were classified as "in-phase" with discharge (Q), while those that occurred after were "out-of-phase" with Q, as proposed by (Van Meter et al. (, 2019). All catchments experienced high stability of the DOC MaxPhase and NO3 MinPhase were the same for all catchments as they always occurred between July and December (Fig. 54, Supplemental S87).

The first type, "in-phase" (68% of the catchments with seasonality), had a $NO_3$ MaxPhase between October and May (Fig. 45, Supplemental S78) (i.e. high-flow period, in-phase with maximum discharge and usually with DOC MinPhase). For these catchments, the mean SI was positive for $NO_3$ (0.22 ± 0.19) and usually negative or null for DOC (0.00 ± 0.13). They tended to be located toward central Brittany and be associated with mesoscale catchments (mean of 52.6 ± 38.8 km²). They had large Ampli for $NO_3$ and low Ampli for DOC (mean relative Ampli of 0.83 ± 0.46, and 0.44 ± 0.23 for DOC) and relatively low C50 of $NO_3$ (means of 5.74 ± 2.46 mg N.l$^{-1}$ and 5.92 ± 2.00 mg C.l$^{-1}$).

The second type, "out-of-phase" (32% of the catchments with seasonality), had a DOC MinPhase and $NO_3$ MaxPhase between May and September (Fig. 45; Supplemental S87) (i.e. low-flow period, out-of-phase with maximum discharge). For most catchments, maximum $NO_3$ and minimum DOC concentrations occurred a mean of 1.85 months before minimum discharge or 5.5 months after maximum discharge, respectively. For these catchments, the mean SI was negative or null for $NO_3$ (-0.08 ± 0.06) and weakly positive for DOC (0.21 ± 0.10). These catchments were close to the coast and relatively small (mean of 31.4 ± 21.7 km²). The had smaller Ampli than "in-phase" catchments for $NO_3$, and higher Ampli for DOC (mean relative Ampli of 0.13 ± 0.13, and 0.74 ± 0.30 for DOC) and relatively high C50 of $NO_3$ (means of 8.27 ± 2.90 mg N.l$^{-1}$ and 5.00 ± 1.62 mg C.l$^{-1}$).

Some catchments had intermediate behavior between these two types (Figs. 45 and 56). Some had a plateau with maximum $NO_3$ and minimum DOC concentrations from winter to summer, while others showed two maxima for $NO_3$ or two minima for DOC (one synchronous with maximum discharge and another with minimum discharge). Other catchments also had maximum $NO_3$ synchronous with discharge, but minimum DOC after maximum discharge.

The seasonal dynamics of SRP were more stable than those of DOC and $NO_3$, but less stable than those of discharge. Thus, there was only one type of seasonality for SRP, which was out-of-phase with flow: MaxPhase SRP dominated in summer (mid-August ± 1.4 months), and MinPhase SRP dominated in late winter (March ± 1.2 months) (Fig. 54, Supplement S76), except for two catchments with maximum SRP in January-February.

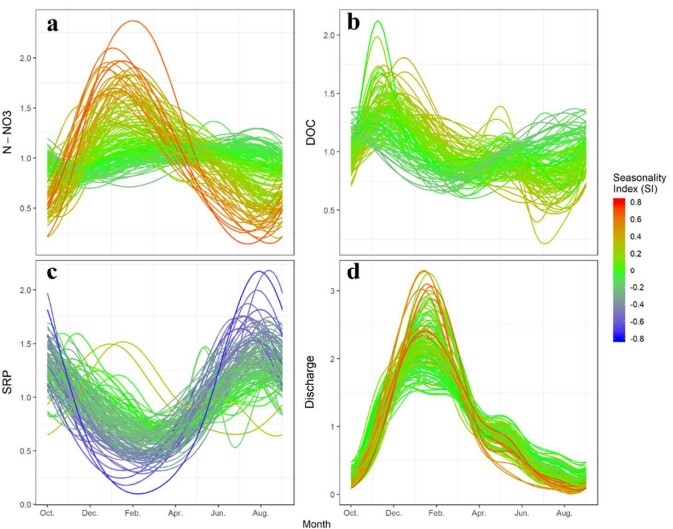

**Figure 5̶4**. Seasonal dynamics of ̶: a) nitrate N (N-NO₃), **b)** dissolved organic carbon (DOC), **c)** soluble reactive phosphorus (SRP), and **d)** daily discharge modeled by Generalized Additive Models, for 185 headwater catchments. To compare concentrations, they are standardized by their mean interannual concentration. The color gradient represents the seasonality index of each parameter; thus, a headwater catchment's color can vary among panels.

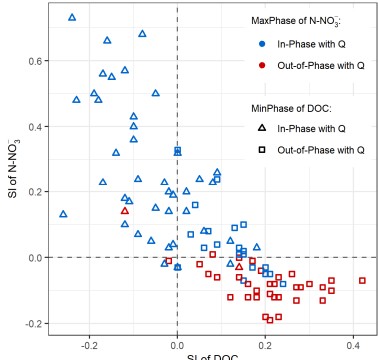

**Figure 6̶5**. Relationship between the seasonality indices (SI) of nitrate N (N-NO₃) vs. dissolved organic carbon (DOC) in the headwater catchments for which seasonality was significant for both parameters (n=98). The color and shape of symbols identify the seasonality types based on the NO₃ MaxPhase and DOC MinPhase metrics. The threshold date was 1 May: MaxPhase that

occurred before were classified as "in-phase" with discharge (Q), while those that occurred after were "out-of-phase" with Q. The DOC MinPhase metric is shown to highlight the synchrony between minimum DOC and maximum N-NO$_3$ concentrations.

**3.3 Controlling factors of concentration and discharge percentiles and seasonality**

The C50 of DOC was correlated significantly with 15 spatial variables and most strongly ($|r_s| \geq 0.4$) with topographic index, QMNA, and the other hydrological indices. The C50 of NO$_3$ was correlated significantly with 12 spatial variables, in particular diffuse agricultural sources ($r_s = 0.68$ for the percentage of summer crops, $r_s > 0.39$ for N and P surplus, and $r_s = 0.48$ for soil erosion rate) and hydrological indices, through the base flow index (BFI) (positively) and W2 (negatively), (Table 3). The C50 of SRP was correlated significantly with more variables (18), but the correlations were slightly weaker. It correlated most strongly with soil P stock ($r_s=-0.40$), climate and hydrology ($r_s=-0.43$ to $-0.34$ with effective rainfall, Qmean, QMNA), elevation, and hydrographic network density. It had weaker positive correlations ($r_s < 0.3$) with the soil erosion rate and domestic and agricultural pressures (urban percentage and P surplus).

Ampli and SI for DOC and NO$_3$ were correlated most with the hydrodynamic properties, followed by agricultural pressures (Fig. 7, Table 3). The catchments "in-phase" with discharge (i.e. positive SI-NO$_3$ and negative SI-DOC correlations) were associated with high hydrological reactivity (low BFI and high W2) and a low percentage of summer crops (Table 3). Conversely, catchments "out-of-phase" with discharge (i.e. negative SI-NO$_3$ and positive SI-DOC correlations) were associated with low hydrological reactivity (high BFI and QMNA, low W2) and a high percentage of summer crops.

Correlations of SI with catchment descriptors were weaker ($|r_s| \leq 0.4$) for SRP than for DOC, and NO$_3$ and discharge because most catchments had the same seasonal pattern, with maximum SRP concentration during low flow. Catchments with the highest amplitudes of SRP concentration were associated with low QMNA and Qmean, high W2, low effective rainfall, and low soil P stock. Interannual loads were correlated mainly with hydrological descriptors (positively with Qmean and QMNA, and negatively with W2) (Table 3). Interannual NO$_3$ loads were also correlated with the percentage of summer crops and soil TP content, while interannual SRP loads were correlated weakly with the percentage of summer crops, agricultural surplus, erosion, and point sources. Discharge indicators present some obvious correlations (e.g. Q50 and annual amplitude with Qmean and QMNA). Q50, and in a lower degree, annual amplitude are positively correlated with baseflow index (BFI), and negatively correlated with flow flashiness (W2). This indicates that in catchments where streams are more influenced by groundwater (generally those flowing on granite), BFI is high and flow flashiness is low.

Correlations with catchment characteristics are lower than expected for the Q50. Q50 is significantly correlated with wetness topographic index (meanTWI, rs = -0.53) which indicates that Q50 is increasing in catchments with drier soils (meanTWI low). Positive correlation with granite indicates that discharge is more supported by this type of rocks, which present favorable groundwater storage. Q50 is positively correlated with soil TP, which is higher on granite substratum. Q50 is positively correlated with SummerCrop and negatively with WinterCrop, underlying higher runoff in catchments with non-cultivated soil during winter.

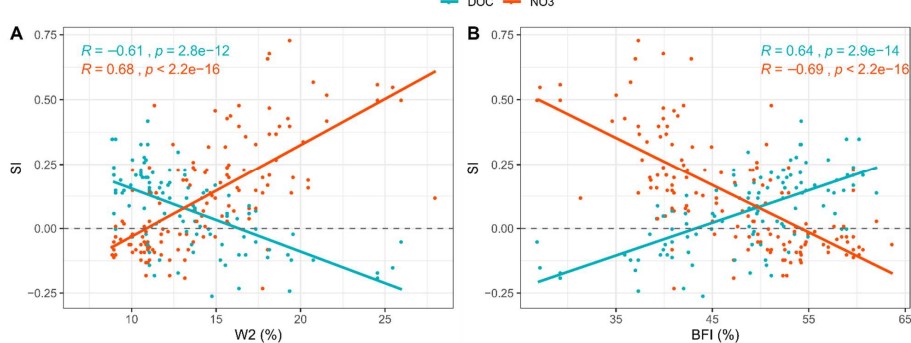

**Figure 67. Relationship between the seasonality index (SI) of dissolved organic carbon (DOC) and nitrate (NO₃) and the hydrological reactivity descriptors (A) flow flashiness index (W2) and (B) base-flow index (BFI) for 124 headwater catchments.**

**Table 3. Spearman rank correlations between water quality indices and geographical descriptors for dissolved organic carbon (DOC), nitrate (NO₃), and soluble reactive phosphorus (SRP). Only significant correlations (p≤0.05) are shown, and bold text indicates |r| ≥ 0.40.**

| | Spatial variable | **DOC** | | | | **NO₃** | | | | **SRP** | | | |
|---|---|---|---|---|---|---|---|---|---|---|---|---|---|
| | | C50 | Ampli | SI | Load | C50 | Ampli | SI | Load | C50 | Ampli | SI | Load |
| **Topography** | Area | – | -0.24 | – | – | – | – | – | – | – | – | – | – |
| | Elevation | **-0.46** | -0.18 | – | – | – | -0.31 | -0.20 | 0.19 | -0.20 | – | – | – |
| | Density_hn | – | – | – | – | – | -0.22 | – | 0.16 | -0.30 | -0.27 | 0.19 | – |
| | Topo_i | **0.54** | – | – | – | – | **0.41** | 0.25 | -0.33 | 0.39 | 0.25 | – | 0.18 |
| | IDPR | – | – | – | – | – | – | – | – | -0.21 | -0.19 | – | – |
| **Geology** | Granite_pm | – | – | 0.21 | **0.41** | – | **-0.43** | -0.31 | 0.27 | -0.26 | -0.24 | – | – |
| | Schist_pm | – | -0.21 | -0.37 | -0.29 | -0.16 | 0.25 | 0.22 | -0.23 | – | – | – | -0.20 |
| | Other_pm | – | 0.32 | 0.35 | – | 0.28 | – | – | – | 0.28 | 0.16 | – | 0.35 |
| **Soil** | Erosion | -0.36 | 0.24 | – | – | **0.48** | 0.16 | -0.26 | 0.39 | 0.24 | 0.17 | – | 0.33 |
| | OC_soil | -0.27 | -0.21 | – | – | – | -0.29 | – | 0.18 | -0.20 | -0.19 | – | – |
| | TP_soil | **-0.44** | – | – | 0.38 | – | **-0.51** | -0.34 | **0.49** | **-0.40** | -0.32 | – | – |
| **Land use** | SummerCrop | -0.30 | 0.28 | **0.54** | – | **0.68** | – | **-0.47** | **0.54** | – | – | 0.29 | 0.36 |
| | WinterCrop | 0.19 | – | -0.20 | -0.29 | – | **0.48** | 0.21 | -0.23 | 0.17 | – | -0.18 | – |
| | Forest | – | -0.17 | -0.30 | 0.23 | -0.37 | **-0.47** | – | – | -0.29 | -0.19 | – | -0.27 |
| | Pasture | – | – | – | – | -0.30 | – | 0.26 | -0.20 | – | – | – | – |
| | Urban | – | – | – | – | – | – | – | – | 0.23 | – | – | – |
| **N and P diffuse and point sources** | N_surplus | -0.21 | 0.20 | – | – | 0.39 | – | – | 0.38 | – | – | 0.29 | 0.29 |
| | P_surplus | -0.24 | 0.33 | – | -0.22 | **0.49** | – | -0.32 | 0.37 | 0.20 | -0.19 | – | 0.35 |
| | N_point | – | -0.17 | – | – | – | – | – | – | – | – | – | – |
| | P_point | – | -0.16 | – | – | – | – | – | 0.21 | – | – | – | 0.21 |
| **Hydrology** | Qmean | **-0.49** | 0.19 | – | **0.53** | 0.16 | **-0.58** | **-0.42** | **0.67** | -0.39 | -0.31 | 0.21 | 0.18 |
| | QMNA | **-0.52** | 0.25 | **0.41** | **0.48** | **0.42** | **-0.54** | **-0.56** | **0.76** | -0.34 | -0.32 | 0.35 | 0.27 |
| | BFI | **-0.41** | -0.27 | **0.64** | 0.38 | **0.54** | **-0.52** | **-0.69** | **0.57** | -0.20 | -0.23 | 0.32 | 0.23 |
| | W2 | **0.43** | – | **-0.61** | **-0.46** | **-0.49** | **0.54** | **0.68** | **-0.59** | 0.20 | 0.20 | -0.26 | -0.24 |
| | Precipitation | **-0.50** | – | – | **0.47** | – | **-0.60** | -0.39 | **0.60** | **-0.43** | -0.33 | 0.18 | – |
| | Wetland | 0.16 | – | 0.31 | 0.38 | – | – | – | – | – | – | – | 0.35 |

**Table 3.** Spearman rank correlations between water quality indices for dissolved organic carbon (DOC), nitrate (NO₃), soluble reactive phosphorus (SRP), discharge indices (Q) and geographical descriptors. Only significant correlations (p≤0.05) are shown, and bold text indicates |r| ≥ 0.40.

| | Spatial variable | DOC C50 | DOC Ampli | DOC SI | DOC Load | NO₃ C50 | NO₃ Ampli | NO₃ SI | NO₃ Load | SRP C50 | SRP Ampli | SRP SI | SRP Load | Q Q50 | Q Ampli |
|---|---|---|---|---|---|---|---|---|---|---|---|---|---|---|---|
| **Topography** | Area | - | -0.24 | - | - | - | - | - | - | - | - | - | - | - | - |
| | Elevation | **-0.46** | -0.18 | - | - | - | -0.31 | -0.20 | 0.19 | -0.20 | - | - | 0.38 | 0.38 | 0.37 |
| | Density_hn | - | - | - | - | - | -0.22 | - | 0.16 | -0.30 | -0.27 | 0.19 | 0.25 | 0.25 | - |
| | meanTWI | **0.54** | - | - | - | - | **0.41** | 0.25 | -0.33 | 0.39 | 0.25 | - | **-0.53** | **-0.53** | **-0.59** |
| | IDPR | - | - | - | - | - | - | - | - | -0.21 | -0.19 | - | 0.20 | 0.20 | - |
| **Geology** | Granite_pm | - | - | 0.21 | **0.41** | - | **-0.43** | -0.31 | 0.27 | -0.26 | -0.24 | - | **0.43** | **0.43** | 0.35 |
| | Schist_pm | - | -0.21 | -0.37 | -0.29 | -0.16 | 0.25 | 0.22 | -0.23 | - | - | - | -0.25 | -0.25 | - |
| | Other_pm | - | 0.32 | 0.35 | - | 0.28 | - | - | - | 0.28 | 0.16 | - | - | - | - |
| **Soil** | Erosion | -0.36 | 0.24 | - | - | **0.48** | 0.16 | -0.26 | 0.39 | 0.24 | 0.17 | - | - | - | - |
| | OC_soil | -0.27 | -0.21 | - | - | - | -0.29 | - | 0.18 | -0.20 | -0.19 | - | 0.34 | 0.34 | 0.32 |
| | TP_soil | **-0.44** | - | - | 0.38 | - | **-0.51** | -0.34 | **0.49** | **-0.40** | -0.32 | - | **0.78** | **0.78** | **0.71** |
| **Land use** | SummerCrop | -0.30 | 0.28 | **0.54** | - | **0.68** | - | **-0.47** | **0.54** | - | - | 0.29 | 0.29 | 0.29 | - |
| | WinterCrop | 0.19 | - | -0.20 | -0.29 | - | **0.48** | 0.21 | -0.23 | 0.17 | - | -0.18 | **-0.51** | **-0.51** | -0.34 |
| | Forest | - | -0.17 | -0.30 | 0.23 | -0.37 | **-0.47** | - | - | -0.29 | -0.19 | - | - | - | 0.25 |
| | Pasture | - | - | - | - | -0.30 | - | 0.26 | -0.20 | - | - | - | - | - | - |
| | Urban | - | - | - | - | - | - | - | - | 0.23 | - | - | - | - | - |
| **N and P diffuse and point sources** | N_surplus | -0.21 | 0.20 | - | - | 0.39 | - | - | 0.38 | - | - | 0.29 | 0.28 | 0.28 | - |
| | P_surplus | -0.24 | 0.33 | - | -0.22 | **0.49** | - | -0.32 | 0.37 | 0.20 | -0.19 | - | 0.20 | 0.20 | - |
| | N_point | - | -0.17 | - | - | - | - | - | - | - | - | - | - | - | - |
| | P_point | - | -0.16 | - | - | - | - | - | 0.21 | - | - | - | - | - | - |
| **Hydrology** | Qmean | **-0.49** | 0.19 | - | **0.53** | 0.16 | **-0.58** | **-0.42** | **0.67** | -0.39 | -0.31 | 0.21 | **0.95** | **0.95** | **0.90** |
| | QMNA | **-0.52** | 0.25 | **0.41** | **0.48** | **0.42** | **-0.54** | **-0.56** | **0.76** | -0.34 | -0.32 | 0.35 | **0.94** | **0.94** | **0.70** |
| | BFI | **-0.41** | -0.27 | **0.64** | 0.38 | **0.54** | **-0.52** | **-0.69** | **0.57** | -0.20 | -0.23 | 0.32 | **0.72** | **0.72** | 0.21 |
| | W2 | **0.43** | - | **-0.61** | **-0.46** | **-0.49** | **0.54** | **0.68** | **-0.59** | 0.20 | 0.20 | -0.26 | **-0.76** | **-0.76** | -0.30 |
| | Precipitation | **-0.50** | - | - | **0.47** | - | **-0.60** | -0.39 | **0.60** | **-0.43** | -0.33 | 0.18 | **0.88** | **0.88** | **0.86** |
| | Wetland | 0.16 | - | 0.31 | 0.38 | - | - | - | - | - | - | - | - | - | - |

**4 Discussion**

**4.1 Interpretation of the spatial opposition between DOC and $NO_3$**

Spatial opposition between DOC and $NO_3$ concentrations has been reported for a wide range of ecosystems. Taylor and Townsend (2010) found a non-linear negative relationship between them for soils, groundwater, surface freshwater, and oceans, from global to local scales, and highlighted that this negative correlation prevails in disturbed ecosystems. Goodale et al. (2005) reported a similar negative correlation among 100 streams in the northeastern USA. Heppell et al. (2017) found that DOC and $NO_3$ concentrations were inversely correlated with the BFI in six reaches of the Hampshire Avon catchment (UK). Our contribution brings an original focus on this relationship in headwater catchments with high domestic and agricultural pressures. Taylor and Townsend (2010) interpreted this spatial opposition as a response of microbial processes (i.e. biomass production, nitrification, and denitrification) to the ratio of ambient $DOC:NO_3$, which controls $NO_3$ export/retention in catchments (see also Goodale et al. (2005)). In semi-natural ecosystems, high but poorly labile soil organic C pools were associated with lower N retention capacity and thus higher N leaching (Evans et al., 2006). Similarly, several studies (e.g. Hedin et al.,1998; Hill et al., 2000) suggested that DOC supply limits in- and near-stream denitrification. In contrast, other studies claimed that N can influence loss of DOC from soils by altering substrate availability or/and microbial processing of soil organic matter (Findlay, 2005; Pregitzer et al., 2004). In our study, C50 were correlated with both BFI and QMNA, positively for $NO_3$ and negatively for DOC, which suggests that catchments strongly sustained by groundwater flow produced higher $NO_3$ and lower DOC concentrations, as reported in other rural catchments (e.g. Heppell et al., 2017). The C50 of $NO_3$ increased with agricultural pressures (percentage of summer crop, N surplus), as observed by Lintern et al. (2018), while that of DOC increased in flatter catchments, which is consistent with results of Mengistu et al. (2014) and Musolff et al. (2018). This suggests that this spatial opposition between DOC and $NO_3$ results from the combination of heterogeneous human inputs, heterogeneous natural pools, and different physical and biogeochemical connections between C and N pools. In surface water, these heterogeneous sources are expressed to differing degrees depending on the catchment's hydrological behavior. When deep or slow flowpaths dominate, they store and release N via groundwater and mobilize little the sources rich in organic matter. When shallower and faster flowpaths dominate, they transport some of the N via compartments rich in organic matter, which causes N depletion and release of more DOC to the streams. The initial amounts of $NO_3$ along these flowpaths are a function of human pressures.

**4.2 Interpretation of the temporal opposition between DOC and $NO_3$**

The seasonal opposition between DOC and $NO_3$ concentration dynamics could be another manifestation of the spatial opposition between DOC and $NO_3$ sources, because the strength of the hydrological connection between sources and streams

varies seasonally (e.g. Mulholland and Hill (1997);, Weigand et al. (2017)). The direct contribution of biogeochemical reactions that connect DOC and $NO_3$ cycles may also vary seasonally (Mulholland and Hill, 1997; Plont et al., 2020). Indeed, temperature, wetness condition, and light availability influence rates of these organic matter reactions (Davidson et al., 2006; Hénault and Germon, 2000; Luo and Zhou, 2006). In addition, the relative importance of the fluxes produced or consumed via these reactions appears clearer during the low-flow period, when the fluxes exported from the terrestrial ecosystem and delivered to the stream decrease. These reactions consume $NO_3$ (e.g. denitrification, biological uptake) and release (reductive dissolution) or produce (autotrophic production) DOC. Of the two seasonal $NO_3$-DOC cycles, the most common in our datasets is thus maximum $NO_3$ in-phase with maximum discharge and minimum DOC, which has been reported in Brittany (Abbott et al., 2018b; Dupas et al., 2018) and elsewhere (Van Meter et al., 2019; Dupas et al., 2017; Halliday et al., 2012; Minaudo et al., 2015; Weigand et al., 2017). The main control of seasonal DOC-$NO_3$ cycles appears to be related to hydrological indices (expressed as BFI and W2). Hydrological flashiness reflects the relative importance of subsurface flow compared to deep base flow (Heppell et al., 2017); thus, low BFI (or high W2) would indicate higher connectivity with subsurface riparian sources and shorter transit times. This is consistent with results of Weigand et al. (2017), who observed higher seasonal amplitudes in DOC and $NO_3$ concentrations and stronger temporal anti-correlation between DOC and $NO_3$ concentrations in stream water dominated by subsurface runoff.

Our results are consistent with these previous results, while the correlations with catchment characteristics can provide some explanation. Catchments with low BFI have larger shallow flows and experience seasonal DOC-$NO_3$ cycles that are in-phase with flow and have higher $NO_3$ amplitudes. These cycles can be interpreted as the combination of several mechanisms (Fig. 8~~7~~):

1) Synchronization, (-i.e, coincident timing). ~~at the same time,~~ of $NO_3$-rich and DOC-poor groundwater contribution with maximum flow.

2) Large contribution of near-/in-stream biogeochemical processes at reduced low flows that decreases $NO_3$ concentration (e.g. $NO_3$ consumption by aquatic microorganisms, biofilms, ~~and~~ macrophytes, and redox processes).

3) Large DOC-rich riparian contribution throughout the year, but larger in autumn, when flow starts to increase, as described in detail in previous AgrHys Observatory studies (Aubert et al., 2013; Humbert et al., 2015).

In contrast, catchments with higher BFI have smaller shallow flows and experience mainly DOC and $NO_3$ cycles that are out-of-phase with flow and have lower amplitudes. These cycles can be attributed to the following:

1) More continuous groundwater contribution, combined with a decrease in agricultural pressures over time, and consequently, a decrease of $NO_3$ concentration in ~~more in ss~~hallower ~~and/~~younger groundwater ~~groundwater~~ than in the deeper/,older ~~groundwater~~one ~~which could increase $NO_3$ concentrations more in deeper groundwater than in shallower groundwater~~ (Abbott et al., 2018b; Martin et al., 2004; Martin et al., 2006). This vertical gradient in groundwater supply could explain why $NO_3$ concentrations peaked during the annual discharge recession, which is sustained mainly by deep groundwater inputs.

2) Little contribution of near-/in-stream biogeochemical processes at reduced low flows due to larger inputs from groundwater, which maintains a relatively high minimum $NO_3$ concentration.

3) Contribution of DOC-rich riparian sources, mainly in autumn, that are smaller than those in in-phase catchments, again due to a predominantly deeper geometry of water circulation.

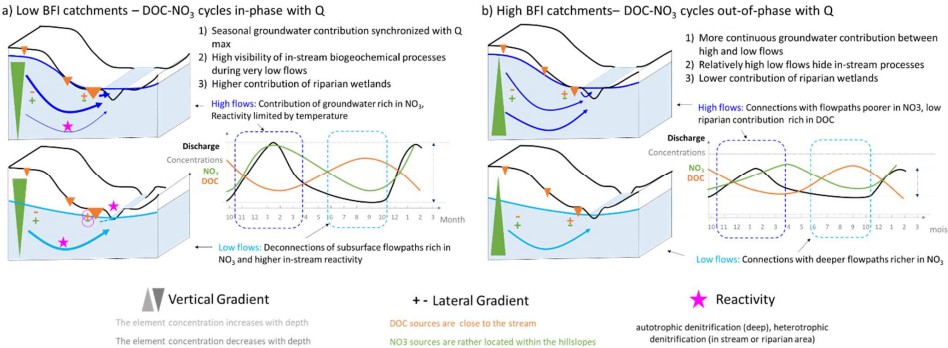

**Figure 87. Conceptual diagram of seasonal flowpaths involved in the DOC-$NO_3$ seasonal cycles leading to a) in-phase cycles with discharge or b) out-of-phase cycles with discharge.**

### 4.3 Interpretation of the spatial and temporal signature of SRP

The correlations between the C50 of SRP and geographic variables highlighted the importance of P sources (soil P stocks,
followed by domestic and agricultural pressures) and surface flowpaths (e.g. hydrological indices, elevation, erosion risk). Similarly, analysis of regression models that predicted spatial variability in total P concentration of 102 rural catchments in Australia also indicated positive effects of human-modified land uses, natural land uses prone to soil erosion, mean P content of soils, and to a lesser extent, topography (Lintern et al., 2018). They always included the percentage of urban area, which suggests a considerable effect of sewage discharge, even at low levels of urbanization. The catchments analyzed in the present
study have a homogeneous and relatively dense distribution of small villages but no large city, which seems to support this last hypothesis. Sobota et al. (2011) studied spatial relationships among P inputs, land cover and mean annual concentrations of different forms of P in 24 catchments in California, USA. They found that P concentrations were significantly correlated with agricultural inputs and, to a lesser extent, agricultural land cover but not with estimates of sewage discharge. Nonpoint sources of P in agricultural runoff, historical inputs of fertilizer and manure in excess of crop requirements have led to a build-
up of soil P levels, particularly in areas of intensive crop and livestock production (Sharpley et al., 1994). This led to correlations between soil P and runoff concentrations in agricultural catchments (Cooper et al., 2015; Sandström et al., 2020), as found here.

The seasonality of SRP was generally the same in the region studied, and C50 and amplitudes were significantly correlated. A peak in seasonal SRP concentrations at low flow has been reported previously (Abbott et al., 2018b; Bowes et al., 2015; Dupas et al., 2018; Melland et al., 2012). It is interpreted as the result of a dominance of point sources diluted during high flow (Minaudo et al., 2019, 2015; Bowes et al., 2011) or of stream-bed sediment sources for which P release increases with temperature (Duan et al., 2012).

Correlation between spatial patterns of $NO_3$ and SRP was expected given the dominant agricultural origin of N and substantial agricultural origin of P, but it was not observed in all catchments. The C50 of $NO_3$ and SRP were high mainly on the northwestern coast, perhaps due to intensive vegetable production associated with a dominance of mineral fertilization (Lemercier et al., 2008). Elsewhere, a high proportion of allochthonous P in the topsoil results from livestock farming and manure application (Delmas et al., 2015). The P-retention capacity of soils (related to their Al, Ca, Fe, and clay contents) is also likely to increase spatial variability in the release of P from catchments (Delmas et al., 2015). Synchronous variations in SRP and DOC, such as those observed in small, completely agricultural headwater catchments without villages (Cooper et al., 2015; Dupas et al., 2015b; Gu et al., 2017), were not observed in the present set of catchments. We assume that synchronicity of SRP and DOC in small catchments depends on soil processes, such as reduction of soil Fe-oxyhydroxides in wetland zones (Gu et al., 2019), which are hidden by in-stream processes (P adsorption on streambed sediments) and downstream point-source inputs (especially P inputs) in the set of larger catchments studied.

Regarding the geographic data used as spatial descriptors, the region studied did not have a few dense urban centers but rather smaller domestic points scattered across the region, which is harder to characterize finely. Moreover, Brittany's coastlines may have higher population densities in spring and summer due to tourism. Refined estimates of domestic point sources and their seasonal variations would be useful in future analyses.

### 4.4 Hydrological vs. anthropogenic controls of spatial variability in water quality

Among the headwater catchments selected, the human pressures (agriculture for $NO_3$ and sewage water discharge for SRP) influenced the C50 and loads of $NO_3$ and SRP. However, the influence of hydrological descriptors on the spatial variability in their loads suggested a transport-limited behavior of these catchments (Basu et al., 2010). Nutrient load estimates had high uncertainties due to i) using modeled flow data when measurements were not available and ii) the frequency of concentration data (monthly), which is low for estimating nutrient loads (especially of P) (Raymond et al., 2013). Thus, these load estimates allowed only their relative spatial variation to be analyzed. Although land-use or agricultural pressure variables, in combination with rainfall and discharge variables, are good predictors of nutrient loads at larger scales (Dupas et al., 2015a; Grizzetti et al., 2005; Preston et al., 2011), the correlations with loads were lower in the set of headwater catchments selected. For $NO_3$, this can be explained by higher spatial variability (CVs) in water fluxes than in concentrations (Table 2), which can explain the dominance of hydrological fluxes in the spatial organization of nutrient loads. Such dominance was found to increase with the level of human pressure in Thompson et al. (2011) for $NO_3$. In this study, such relationship was not visible as all the catchments

exhibited a transport-limited behavior. It may also suggest that the nutrient-surplus data at the local scale remained uncertain (Poisvert et al., 2017) or that at this scale, data on agricultural practices would be more relevant, and that variability in concentration depends less on the magnitude of nutrient inputs than on their locations.

The catchments studied have clear seasonal dynamics in concentration, which is consist with previous observations (Minaudo et al., 2019; Abbott et al., 2018a). The seasonal pattern is controlled mainly by hydrological variables. It partly reflects the mixing of contrasting sources that are connected to streams by seasonally varying flowpaths with nutrients that are transferred vs. nutrients that are processed locally in hotspots (e.g. riparian buffer, stream water, stream sediments) or delivered over point sources. The seasonal $NO_3$-DOC pattern seemed to become somewhat homogenous among catchments larger than 100 $km^2$, where seasonal cycles with maximum $NO_3$ in-phase with flow seemed less common. This may be related to an increase in in-stream biological activity during summer as catchment size increases, enhanced by a lower stream water level and slower discharge (Minaudo et al., 2015). Therefore, the potential relationship between seasonal cycle type and catchment size should be studied over a wider range of catchment sizes and nested catchments to include variations along the hydrographic network.

**4.5 Implications for headwater monitoring and management**

The high regional and seasonal variations of nutrient concentrations in streams probably drive high variations of nutrient stoichiometry along the ~~water year~~hydrological cycle and over the region, and, consequently, high variations in time and space of eutrophication risks downstream (Westphal et al., 2020). Due to the combination of anthropogenic and hydrological drivers in explaining these stream concentrations, a better estimation of~~n~~ nutrient inputs and discharge in all headwater catchments is important to predict areas at risks, as a first step~~, is important to predict areas at risks~~. The spatial analysis shows high and poorly structured spatial variations of concentrations over the region. Nevertheless, the opposition between $NO_3$ and DOC concentrations suggests that the C:N ratios will be even more variable:

1) In space: catchments with high DOC C50 and low $NO_3$ C50 will exhibit very high C:N and vice versa

2) Over the seasons: as minimum of DOC and maximum of $NO_3$ concentrations are in-phase: catchment where DOC-$NO_3$ variations are in phase with Q will exhibit a low C:N ratio in winter high flow period and higher C:N ratio during low flow period. The N:P ratio in these catchments will be high during the low flow periods (high $NO_3$ and low SRP concentrations). Catchments where DOC-$NO_3$ variations are out-of-phase with discharge will exhibit probably less variation in their ratios (because of lower $NO_3$ amplitude) with relatively higher winter C:N ratio than the previous type of catchments.

We can stress that monitoring C-N and P is important as each of these elements can follow a different pattern, even in neighboring catchments. Yet, these three basic elements are not always included in water quality monitoring. Therefore, sampling programs in which all three of those elements are quantified should be maintained over the long-term. Such programs

will be necessary to further investigate the variations of these element concentrations in relation with geomorphological and climate conditions.

In this paper, we used inter-annual mean values for DOC, $NO_3$ and SRP loads to establish the spatial variability and seasonal patterns across headwater catchments. Because we demonstrated that the seasonality index (SI) and flow flashiness ($W_2$) are linked, our results can be used to classify non-monitored catchments as a function of their potential load flashiness. Flow flashiness ($W_2$) combined with SI, or the slope of C-Q relationships for high flows, could be employed for a sampling or monitoring design to improve annual or seasonal load estimations for the most contributive catchments (Moatar et al, 2020). However, other issues, such as the assessment of eutrophication risk for some lakes, estuaries or bays around the peninsula would require more frequent sampling, especially for SRP.

## 5 Conclusion

To analyze spatial variability in water quality at a regional scale, we used an original dataset from public databases, little used by the scientific community, for the French region of Brittany with monthly measurements of water quality. The dataset selected covers 185 headwater and agricultural catchments monitored over a period sufficiently long (10 years) to allow the spatial (regional) variability and temporal (seasonal) variation in DOC, $NO_3$, and SRP concentrations to be analyzed. We described spatio-temporal variations in concentrations, loads, and seasonal patterns and analyzed their correlations with geographic variables (related to topography, hydro-climate, geology, soils, land uses, and human pressures). Our study showed the following:

1) Seasonal cycles of DOC and $NO_3$ concentrations are usually opposite from each other. Catchments with a low base-flow index exhibit maximum $NO_3$ in-phase with maximum flow, while those with a higher base-flow index exhibit maximum $NO_3$ after maximum flow. Both types exhibited maximum DOC in autumn, at the beginning of the annual increase in flow.

2) $NO_3$ concentrations increased as human pressures and base flow contribution increased. DOC concentrations decreased as rainfall, base flow contribution, and elevation increased. SRP concentrations showed weaker correlations with human pressures, rainfall, and hydrological and topographic variables.

3) Seasonal SRP cycles are synchronized in nearly all catchments that have a clear seasonal amplitude, with maximum SRP concentrations that occur during the summer low-flow period due to a decreased dilution capacity of point sources.

The spatial and temporal opposition between DOC and $NO_3$ concentrations likely results from a combination of heterogeneous human inputs and biogeochemical connection between these pools. The seasonal cycles in stream concentrations result from the mixing of water parcels that followed contrasting flowpaths, combined with high spatial variability in nutrient sources,

local-scale biogeochemical processes, and point sources. As a perspective, we recommend further studies of multiple elements that are likely to show contrasting responses to diverse human pressures and to the retention/removal capacities of hydrosystems.

**Acknowledgments**

The salary of SG was supported by Region Bretagne and Agence de l'Eau Loire Bretagne. We thanks Dr Remi Dupas (INRAE Rennes) for his valuable contribution for methodological choices and the scientific interpretations and discussions. We thank Dr Vazken Andreassian (INRAE Anthony) for providing regional simulations of discharge time series with the model GR4J. We thank also Josette Launay (CRESEB), Elodie Bardon (Observatoire Environnement Bretagne), Yves-Marie Heno and Olivier Nauleau (DREAL Bretagne) for their contributions to the data selection and to the project. Finally, we thanks all the people who contributed to the collection of public data on surface water quality in French Brittany.

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
