# Peer review of "Spatio-temporal controls of C-N-P dynamics across headwater catchments of a temperate agricultural region from public data analysis"

_Hydrology and Earth System Sciences, 2020_

## Referee Comment (RC1) · Anonymous Referee #1 · 18 Aug 2020

General comments

This study jointly explores concentrations and concentration seasonality of nitrate, DOC and phosphate in the agricultural Brittany area, France. It provides an impressive database with surprisingly strong hydroclimatic gradients. Using different statistical methods the study try to explain the interplay human impact and given catchment conditions shaping the observed responses. This all is interesting for catchment scientists and water quality managers and suitable for HESS. While I like the general approach of the study I did not find it totally convincing. I see issues with the applied methodology as well as with the way the results are discussed. In terms of methods I don't clearly

see the added value of the PCA and the GAM. Statements made from the PCA could have been made from simple correlation analysis as well. The GAM selects only catchments with a significant seasonality and discards chemostatic catchments. The basic findings could have maybe been also derived by simply describing seasonality indices and/ or a averaging of concentrations for each month of the year. Finally, the correlation analysis with the catchment variables should touch and discuss covariation among the predicting variables. This often hinders interpretation towards underlying processes. Concerning the discussion of the result I miss a synthesis that goes beyond the mere description of the patterns and the statement of interacting natural and anthropogenic drivers. What are implications for ecological water quality (stoichiometry!). What are implications for management and potential future development of these catchments? How does that embed into other research that address catchment scale water quality? This all does not have to be exhausting but would push the paper away from the more descriptive style to something that adds more scientific value. For details see my specific comments below.

Specific comments

Abstract

I would have expected some discussion part on the underlying processes here. You describe patterns but you do not discuss these. Why?

L23: "opposing pattern" would maybe fits better here.

Introduction

L39: Mentioning headwater catchments here seems to be disconnected from the line of argumentation. Why is it relevant to look at headwaters? You mention that later - maybe start with that argument here.

L49: Other studies such as Zarnetske et al. (2018, 10.1029/2018gl080005) or Musolff et al. (2018, 10.1016/j.jhydrol.2018.09.011) indicate a dominance of topography and

connected wetlands in terms of concentrations (not DOC quality).

L45-54: This exploration of human impacts on C, N and P concentration and spatial concentration variability is not totally convincing. I think some more words, a clear structure and a systematic evaluation of all three nutrients is needed. I miss a discussion on the spatial homogenization by agriculture that was discussed by Basu et al. (2010, 10.1029/2010gl045168) and Basu et al. (2011, 10.1029/2011wr010800).

L69: Why need the human pressure to be similar in headwater catchments to study them better?

L72: The reference (Agren) here has an unclear meaning. Does this study state the lack of seasonal analysis or also do not consider seasonality or consider as a rare case seasonality?

L78f: This hypotheses needs to be better worked out above - see my comment above (referring to L45-54).

L84: What are "relevant" time series?

L87: I suggest to leave out "potential" here. The causality of the correlation may be potentially hint to an underlying process.

Material and Methods

Table 1: Catchment descriptors are not always self-explaining: What is the topographic index? Is elevation referring to the mean elevation? What is the "class" of dominant soil thickness?

eq 1: Did you considered the offset when the discharge gauge was not at the same position as the water quality station?

L172ff: Did I rightly understood that GAM considered month of the year as only variable? This is not fully clear from the text. Later on it looks like day of the year was the predicting variable.

L177: "Amplitude" of a trend is maybe not the right wording. "Slope" is totally fine.

L179: I don't understand this last sentence.

Results

L212f: This is already a discussion of your result and should thus be part of the discussion section.

L213ff: All these statements could have been made from a correlation analysis of C10, C50 and C90 (among and between the three nutrients) only. I do not see the added value of the PCA - from my point of view it may be taken out.

L231f: Check this sentence. Better "fitted to XX DOC concentration time series"?

L232f: Can you quantify that? Is mean SI lower for the cases where GAM could not be fitted?

L241: Check this sentence. Discharge cannot have a seasonal concentration cycle.

L244: Does that refer to the comparison between all catchments? That is not clear here.

L245f: I am not sure were to see this gradient in Fig. 4. Is that referring to the right figure?

L257f: What does that stability means? That the pattern does not change between the years? This cannot be seen from the GAM averaging over all years. I am a bit lost here.

L288: You may give direction of the correlation with the hydrologic variables as well.

Discussion

L304ff: Rather than directly with the interaction of N and C wouldn't it be better to first explain the individual spatial patterns?

L313ff: But this argument would lead to high concentrations of both, C and N?

L324ff: Wouldn't Fovet et al. (2018, 10.1016/j.jhydrol.2018.02.040) provide a good mechanistical backup for the processes described here?

L334ff: You need some references for these statements.

L350ff: The study may benefit from a conceptual sketch of the two general types of catchments, its N and C sources and seasonal changes.

L400f: You may show and quantify this earlier on by the ratio of CVc and CVq as done in Thompson et al. (2011, 10.1029/2010wr009605).

Conclusions

The conclusions restate the major findings, which is ok for me, but miss implications (e.g. for management) and an overarching synthesis on catchments functioning (in concert with previous studies on e.g. denitrification or solute mobilization from the Brittany [Kolbe et al. 2019, 10.1073/pnas.1816892116], the above mentioned Fovet et al. 2018).

SI

Fig. S1: Panel b does not make sense without a legend. Typo in panel d legend name.

---

## Referee Comment (RC2) · Anonymous Referee #2 · 12 Sep 2020

General Comments: This manuscript provides a descriptive analysis of spatial and temporal patterns of DOC, NO3, and SRP in catchments of the Brittany region of France. The multi-element, many sight approach utilized does provide interesting insight into the potential influences of changing seasonal hydrology/ flowpath and landscape characteristics on the biogeochemistry of the study region. Overall, the manuscript is well written, I enjoyed reading the manuscript, and for the most part the author's interpretation of the patterns observed are supported by their statistical analysis. The paucity of other studies focusing on multi-element patterns, in headwater streams, that examine seasonal patterns, or that focus on multiple catchments is somewhat overemphasized in the framing of the research though and further cross

comparison with studies that include all or only some of those criteria would benefit the introduction and discussion. In specific comments a number of potentially helpful references to similar research are noted. Regarding the GAM model used to describe seasonality, this is a useful approach, but I also wonder if there may be opportunity to modify the presentation and possibly the models slightly to explore interactions between multiple drivers (e.g. season x land use or flow x soil).

Specific comments: Lines 45-50 – There have been a number of studies in Canada and United States to evaluate the influence of agricultural land use on DOC concentration and DOM composition. Although the statement that composition is usually quite altered is true, often concentration is more a function of the same factors as in non-agricultural catchments, in particular the presence of wetlands and soil drainage properties.

Lines 65-75- I understand the point that the authors are making here, but there are actually a number of studies meeting most of these criteria that could be helpful in interpretation of results and in understanding the generality of the patterns observed across regions. A couple of ideas that came to mind when reading this section were:

Fasching et al. 2019 in Ecosystems also use GAM models and the approach used to explore multiple drivers may be helpful, Natural land cover in agricultural catchments alters flood effects on DOM composition and decreases nutrient levels in streams - https://doi.org/10.1007/s10021-019-00354-0

Although larger watersheds in the region are also included in the analysis I would suggest that some comparison should be made with Moatar et al. 2017, WRR, Elemental properties, hydrology, and biology interact to shape concentration–discharge curves for carbon, nutrients, sediment, and major ions https://doi.org/10.1002/2016WR019635

The review and conceptual paper presented by Kaushal et al. 2018 in Biogeochemistry may also be helpful in evaluating the role of season and land use on multi-element water chemistry.

[Figure]

Line 68 – This is true, but there is a lot of study that goes on further upstream in even smaller catchments where land management can be linked directly to impact.

Line 73- maybe also ad "multi-element" to this statement because there are many studies that examine multi-catchment patterns for a single element.

Line 109- This is good. Often selecting sites in a stream network without spatial independence is a pitfall for many site studies in a region, particularly when working with data where the authors did not chose the original sampling locations.

Line 111- Please explain why these criteria were used for outlier selection and how commonly extremely high concentrations were observed.

109-112 – Were data examined to ensure that there were not seasonal biases in the timing of missing data and that certain sites were not heavily sampled only in one season (summer samples only for example)

Line 185- The seasonality metric is interesting, but doesn't really separate the impact of flow condition or discharge from other factors like temperature that vary seasonally. Calculation of a similar metric for high flow vs. low flow for comparison to the SI might be quite revealing. An example of that methods is in Fasching et al. 2019.

Figure 4 – I think the information displayed here is valuable, but I wonder if a visual with additional information might be possible with the GAM results if the influence of 2 different drivers were displayed in a 3d version of the figure similar to Figure 7 in Fasching et al. 2019. It could be discharge or land use on the other axis.

-The discussion on DOC/NO3 patterns is well written and I agree with the authors general interpretation of the results.

-For the SRP discussion it may be worthwhile to reference the strong correlations that have been observed in small agricultural catchments between soil P and runoff concentrations. There are metrics included in the predictor dataset for TP_soil and P_surplus which appear to be model outputs. It may help with interpretation of results if it can

be noted whether these follow anticipated patterns of buildup where more intensive livestock or fertilizer input is occurring.

Line 380 – In the context of the observed seasonal pattern can you comment on the timing of nutrient applications and whether there is potential for depletion of soluble sources over time or not.

Table 1 – Presumably some fields are used for both summer and winter crops. A total % cropland variable might be useful if not already considered.

---

## Author Comment (AC1) · 21 Oct 2020

We appreciated that the Referee #1 found the study "interesting for catchment scientists and water quality managers and suitable for HESS".

He/she raised three major criticisms:

1) Methods: clarifying the choice of PCA and GAM, and analyzing the covariation among predicting variables

1a) Statements made from the PCA could have been made from simple correlation analysis as well. And detailed comment on L213ff: All these statements could have

been made from a correlation analysis of C10, C50 and C90 (among and between the three nutrients) only. I do not see the added value of the PCA - from my point of view it may be taken out.

We agree that correlation coefficients (see table below) led to the same conclusion. The PCA was chosen for graphical representation of the relationships between C10, C50 and C90. Figure S3a was already provided in supplemental rather than in the main text, therefore we suggest adding the correlation coefficients values in the main manuscript to clarify potential questions: "First, percentiles (C10, C50, or C90) were grouped by solute, showing that the spatial organization remained the same regardless of the concentration percentile (Spearman rank correlations between the three indices always greater than 0.56 for all elements). [. . .]. Second, there was a negative correlation between DOC and NO3 concentrations (rs= -0.58; Supplement S3b). Third, SRP concentrations had an orthogonal relation compared to DOC and NO3 concentrations (rs close to zero)."

see Table R1: Spearman's rank correlations between the C10, C50 and C90 metrics for each element

1b) The GAM selects only catchments with a significant seasonality and discards chemostatic catchments. The basic findings could have maybe been also derived by simply describing seasonality indices and/ or a averaging of concentrations for each month of the year.

And detailed comment on L232f: Can you quantify that? Is mean SI lower for the cases where GAM could not be fitted?

GAMs cannot be fitted with reasonable performance if there is no seasonal signal on the time series, thus it does allow for identifying "chemostatic" or - more consistently with the terminology proposed by Van Meter et al. (2019) that we are using in the text - "aseasonal" catchments. The seasonality metrics are then computed from the GAM outputs. For "aseasonal" catchments, amplitude and seasonal index are zero

indeed, whereas PhaseMin and Phase Max cannot be identified (using GAM or not). We agree that several methods can be used to characterize seasonality: averaging concentration (or discharge) of each month through the years is one of them. Here, we chose to smooth the data with a GAM model to limit the influence of outliers and to deal with data gaps: the results eventually look "smoother" than with a monthly aggregation method.

1c) Finally, the correlation analysis with the catchment variables should touch and discuss covariation among the predicting variables. This often hinders interpretation towards underlying processes.

There are indeed correlations among the predicting variables, which are expected, e.g. BFI and W2 are anti-correlated. We suggest adding the correlation matrix Fig R1 in Supplemental.

2) Discussion: a synthesis that goes beyond the description is missing, in regards with previous literature on natural versus anthropogenic drivers

And detailed comments in introduction:

L45-54: This exploration of human impacts on C, N and P concentration and spatial concentration variability is not totally convincing. I think some more words, a clear structure and a systematic evaluation of all three nutrients is needed. I miss a discussion on the spatial homogenization by agriculture that was discussed by Basu et al.(2010, 10.1029/2010gl045168) and Basu et al. (2011, 10.1029/2011wr010800)

The paragraph L45-54 aims at reviewing the reported factors of spatial variability in concentrations among various contexts. The following paragraph L. 55-65 aims at reviewing reported temporal variability in these C, N and P concentrations at the seasonal scale.

There is a considerable literature on the emergence of a chemostatic behavior in catchments due to management and agriculture (Basu et al., 2010; 2011; Thompson et

al., 2011; Musolff et al., 2015; Moatar et al., 2017). Chemostaticity, or biogeochemical stationarity, is defined as the lower variability in water concentration relatively with flow variability (Thompson et al., 2011), so that solute mobilization rates only depends on water fluxes (Basu et al., 2011) and the transport of this solutes is qualified as "transport-limited" (Basu et al., 2010). This chemostaticity is supposed to be the typical behavior of catchments for geogenic solutes because of the geological legacy of "large, ubiquitous source mass distributed within the catchment". In less impacted catchments, the export behavior is expected to be rather source limited as the contemporary sources are distributed within the catchment and because the biogeochemical processes (sorption, degradation) control the amount of solute available for export. These studies hypothesize that, in managed catchments, accumulation of nutrients lead to anthropogenic and spatially homogeneous legacy storages of nutrients within the catchment responsible for the emergence of a chemostatic behavior for these nutrients.

The chemostaticity is determined through the analysis of concentration-discharge or load-discharge relationships or of coefficient variation ratios of concentration versus discharge. It refers rather to the temporal variability of concentration in streams, and usually at inter-annual or long-term scales at which the legacy storages may be viewed as homogeneous within the catchment considering that every year these storages are connected at least during high flow periods (Moatar et al., 2017). Here, we focused on seasonal concentration patterns and they are sensitive to the source spatial distribution within the catchment because of the difference in their connectivity between high and low flow periods. Therefore, the spatial variability in those seasonal patterns does not depend on the management level but rather on the catchment intrinsic properties (topography, geology, climate...)

We suggest adding pieces of discussion to position our study in regards to these published results in the introduction:

"Besides being spatially variable, C, N, and P concentrations also vary temporally.

The variability of concentrations with flow has been described in several studies using concentration-flow relationships at event (Fasching et al., 2019) or inter-annual to long-term scales (Basu et al., 2010; 2011; Moatar et al., 2017). Concentrations also vary seasonally in streams and rivers (Aubert et al., 2013; Dawson et al., 2008; Duncan et al., 2015; Exner-Kittridge et al., 2016; Lambert et al., 2013), as does the composition of dissolved organic matter (Griffiths et al., 2011; Gücker et al., 2016)."

and in the discussion subsection 4.4.:

"For NO3, this can be explained by higher spatial variability (CVs) in water fluxes than in concentrations (Table 2), which can explain the dominance of hydrological fluxes in the spatial organization of nutrient loads. Such dominance was found to increase with the level of human pressure in Thompson et al. (2011) for NO3. In this study, such relationship was not visible as all the catchments exhibited a transport-limited behavior. It may also suggest that the nutrient-surplus data at the local scale remained uncertain (Poisvert et al., 2017) ..."

Fasching, C., et al. (2019). "Natural Land Cover in Agricultural Catchments Alters Flood Effects on DOM Composition and Decreases Nutrient Levels in Streams." Ecosystems 22(7): 1530-1545.

Moatar, F., et al. (2017). "Elemental properties, hydrology, and biology interact to shape concentration-discharge curves for carbon, nutrients, sediment, and major ions." Water Resources Research 53(2): 1270-1287.

Thompson, S. E., et al. (2011). "Relative dominance of hydrologic versus biogeochemical factors on solute export across impact gradients." Water Resources Research 47(10).

Basu, N. B., et al. (2010). "Nutrient loads exported from managed catchments reveal emergent biogeochemical stationarity." Geophysical Research Letters 37(23).

Basu, N. B., et al. (2011). "Hydrologic and biogeochemical functioning of intensively

managed catchments: A synthesis of top-down analyses." Water Resources Research 47(10).

3) Perspective: what are implications for ecological water quality and for management and potential future development of these catchments?

+ detailed comment on the "conclusion" : The conclusions restate the major findings, which is ok for me, but miss implications (e.g. for management) and an overarching synthesis on catchments functioning (in concert with previous studies on e.g. denitrification or solute mobilization from the Brittany [Kolbe et al. 2019, 10.1073/pnas.1816892116], the above mentioned Fovet et al. 2018).

We agree that adding perspectives on ecological and management implications would increase the impact of our article and we suggest adding the following subsection to the discussion section to enlarge these perspectives:

"5.4. Implications for headwater monitoring and management

The high regional and seasonal variations of nutrient concentrations in streams probably drive high variations of nutrient stoichiometry along the water year and over the region, and, as a consequence, high variations in time and space of eutrophication risks downstream (Westphal et al., 2020). Due to the combination of anthropogenic and hydrological drivers in explaining these stream concentrations, a better estimation on nutrient inputs and discharge in all headwater catchments, as a first step, is important to predict areas at risks. The spatial analysis shows high and poorly structured spatial variations of concentrations over the region. Nevertheless, the opposition between NO3 and DOC concentrations suggests that the C:N ratios will be even more variable:

• In space: catchments with high DOC C50 and low NO3 C50 will exhibit very high C:N and vice versa

• Over the season: as minimum of DOC and maximum of NO3 concentrations are

in-phase: catchment where DOC-NO3 variations are in phase with Q will exhibit a low C:N ratio in winter high flow period and higher C:N ratio during low flow period. The N:P ratio in these catchments will be high during the low flow periods (high NO3 and low SRP concentrations). Catchments where DOC-NO3 variations are out-of-phase with discharge will exhibit probably less variation in their ratios (because of lower NO3 amplitude) with relatively higher winter C:N ratio than the previous type of catchments."

Westphal, K., Musolff, A., Graeber, D., and Borchardt, D.: Controls of point and diffuse sources lowered riverine nutrient concentrations asynchronously, thereby warping molar N:P ratios, Environ. Res. Lett., 15, 104009, 2020.

Moreover, to make the link between the interpretations we propose in the discussion and the cited previous studies in similar sites (Kolbe et al., 2019 and Fovet et al., 2018) and following the detailed comment on L350ff "The study may benefit from a conceptual sketch of the two general types of catchments, its N and C sources and seasonal changes.", we suggest adding Figure R2 to illustrate section 4.2.

Reply to specific comments

Abstract: I would have expected some discussion part on the underlying processes here. You describe patterns but you do not discuss these. Why?

We suggest adding two sentences for describing the discussed interpretations of these seasonal cycles in the abstract: "The annual maximum NO3 concentration was in-phase with maximum flow when the base flow index was low, but this synchrony disappeared when flow flashiness was lower. These DOC-NO3 seasonal cycle types were related to the mixing of flowpaths combined with the spatial variability of their respective sources and to local biogeochemical processes. The annual maximum SRP concentration occurred during the low-flow period in nearly all catchments. This likely resulted from the dominance of P point sources. "

L23: "opposing pattern" would maybe fits better here.

The adjective "opposite" refers well here to "inverse" whereas the first sense of "opposing" would be "adverse", while its second definition is indeed "opposite". Then and after crosschecking with an American English native speaker, it seems that the initial formulation was correct.

Introduction

L39: Mentioning headwater catchments here seems to be disconnected from the line of argumentation. Why is it relevant to look at headwaters? You mention that later -maybe start with that argument here. The paragraph from line 39 to line 44 describes why it is rare but relevant to look at headwaters quality. Because focusing on headwaters is a specificity of our study, we found important to explain this point as an element of context before the review and analysis of literature on spatial and seasonal variability of stream water C, N and P concentrations. However to better reconnect this paragraph with the previous we can rephrase as: "In addition, the quality of headwater catchments have been studied less than large rivers (Bishop et al., 2008), despite their influence on downstream water quality and higher spatial variability in their concentrations (Abbott et al., 2018a; Temnerud and Bishop, 2005)."

L49: Other studies such as Zarnetske et al. (2018, 10.1029/2018gl080005) or Musolff et al. (2018, 10.1016/j.jhydrol.2018.09.011) indicate a dominance of topography and connected wetlands in terms of concentrations (not DOC quality).

Indeed, and we describe this observation in the previous sentence (line 45-47): "DOC concentration in streams has been related to topography, wetland coverage, and soil properties such as clay content or pH (Andersson and Nyberg, 2008; Brooks et al., 1999; Creed et al., 2008; Hytteborn et al., 2015; Temnerud and Bishop, 2005).". We suggest adding these two suggested additional references to the citation list line 47.

L69: Why need the human pressure to be similar in headwater catchments to study them better?

Water chemistry in headwater catchments is influenced by human pressure and the catchments' intrinsic buffering capacity. It is easier to disentangle the effect of both factors when one is relatively constant while the other is spatially variable. Several authors demonstrated that Human activities disturbed water quality using catchments depicting a gradient of human pressure. Along a gradient where the percentage of agricultural area varies from 0 to 50 or 60%, with an equilibrate distribution, it is likely that the main driver of spatial variability in water quality (e.g. in NO3 concentration) will be the percentage of agricultural area. Along a gradient where the percentage of agricultural area varies from 60 to 90%, it is likely that other drivers will play a major role in controlling spatial variability of the water quality.

L72: The reference (Agren) here has an unclear meaning. Does this study state the lack of seasonal analysis or also do not consider seasonality or consider as a rare case seasonality?

In Agren et al. (2007), the authors analyzed the importance of seasonality and small streams for regulation of DOC export studying 15 subcatchments (<30 km2) over 3 years. They highlighted that the geographic controls of the spatial variation in DOC exports varied between seasons. We suggest to reformulate this point and change the reference for a list of citations that report seasonal patterns in C, N and/or P stream concentrations: "with little or no analysis of seasonal patterns despite their frequent occurrence (Van Meter et al., 2019; Abbott et al., 2018b; Liu et al., 2014; Halliday et al., 2012; Mullholland et al. 1997)".

L78f: This hypotheses needs to be better worked out above - see my comment above (referring to L45-54).

We suggest adding several references explaining where these hypotheses originate:

"We hypothesized that: 1) Human (i.e. rural and urban) pressures determine spatial variability in NO3 and SRP concentrations (Preston et al., 2011; Melland et al., 2012; Dupas et al., 2015; Kaushal et al., 2018), while soil and climate characteristics deternone

mine that in DOC and possibly SRP (Lambert et al., 2011; Humbert et al., 2015; Gu et al., 2017)."

Preston, S. D., et al. (2011). "Factors Affecting Stream Nutrient Loads: A Synthesis of Regional SPARROW Model Results for the Continental United States1." JAWRA Journal of the American Water Resources Association 47(5): 891-915.

Melland, A. R., et al. (2012). "Stream water quality in intensive cereal cropping catchments with regulated nutrient management." Environmental Science & Policy 24: 58-70.

Dupas, R., et al. (2015). "Assessing the impact of agricultural pressures on N and P loads and eutrophication risk." Ecological Indicators 48: 396-407.

Kaushal, S. S., et al. (2018). "Watershed 'chemical cocktails': forming novel elemental combinations in Anthropocene fresh waters." Biogeochemistry 141(3): 281-305.

Lambert, T., et al. (2013). "Hydrologically driven seasonal changes in the sources and production mechanisms of dissolved organic carbon in a small lowland catchment." Water Resources Research 49(9): 5792-5803.

Humbert, G., et al. (2015). "Dry-season length and runoff control annual variability in stream DOC dynamics in a small, shallowgroundwater-dominated agricultural watershed." Water Resources Research 51(10): 7860-7877.

Gu, S., et al. (2017). "Release of dissolved phosphorus from riparian wetlands: Evidence for complex interactions among hydroclimate variability, topography and soil properties." Science of The Total Environment 598: 421-431.

L84: What are "relevant" time series?

The relevance of the time series refers here to the end of the sentence, i.e. the availability of the four parameters (Q, DOC, NO3, SRP) over a long-term period (10 years) and at medium frequency (monthly).

[Figure]

L87: I suggest to leave out "potential" here. The causality of the correlation may be potentially hint to an underlying process.

We agree with the suggestion.

Material and Methods

Table 1: Catchment descriptors are not always self-explaining: What is the topographic index? Is elevation referring to the mean elevation? What is the "class" of dominant soil thickness?

-The downstream topographic index (Topo_i) is a steady state wetness index commonly used to quantify topographic control on hydrological processes and developed by (Beven and Kirkby, 1979) :

Topo_i=log $\alpha$/tan⁡$\beta$

Where $\alpha$ is the drainage area (ha) and $\beta$ is the downstream slope (%) (Merot et al., 2003). It can be used to predict the spatial distribution of soil wetness: a low Topo_i indicates potentially wet area while a high Topo_i indicates well-drained area.

Beven, K. J. and Kirkby, M. J. (1979) A physically based, variable contributing area model of basin hydrology, Hydrological Sciences Bulletin, 24:1, 43-69, DOI: 10.1080/02626667909491834.

Merot, P., Squividant, H., Aurousseau, P., Hefting, M., Burt, T., Maitre, V., Kruk, M., Butturini, A., Thenail, C., and Viaud, V.: Testing a climato-topographic index for predicting wetlands distribution along an European climate gradient, Ecological Modelling, 163, 51-71, 2003.

- Elevation is the mean elevation of the catchment indeed

- The "dominant soil thickness" classes are 40-60 cm, 60-80 cm, 80-100cm and >100cm.

We agree the information has to be added to Table 1 for the sake of clarity.

eq 1: Did you considered the offset when the discharge gauge was not at the same position as the water quality station?

Yes, we considered the offset when the discharge gauge was not at the same position as the water quality station. When the discharge gauge was not at the same position as the water quality station, the daily flows were extrapolated to the water quality station by multiplying the flow rate by the ratio between the drained areas of the water quality station and the discharge gauge.

L172ff: Did I rightly understood that GAM considered month of the year as only variable? This is not fully clear from the text. Later on it looks like day of the year was the predicting variable.

All GAM for concentrations are obtained by fitting smooth spline functions of month of the year to observed monthly time series. Then, we extracted the values of the fitted GAM at a daily time step. These allowed us to calculate the Cwinter and Csummer, and the SI.

We agree with referee #1 that sentence line 189 introduces some confusion then we suggest rephrasing as: "where Cwinter and Csummer are the averages of winter and summer concentrations, (calculated from daily values from fitted GAM) Âż.

L177: "Amplitude" of a trend is maybe not the right wording. "Slope" is totally fine.

"Amplitude" line 177 refers well to the seasonal amplitude but indeed to avoid th confusion we should modify "amplitude" by "slope" line 176: "First, significant long-term trends (according175to Man-Kendall tests) had low slopes: mean Theil-Sen slopes ranged from -3%to 0% of the median concentration (while mean seasonal relative amplitudes exceeded 50%). "

L179: I don't understand this last sentence.

We suggest rephrasing as "we considered a seasonal dynamic to exist when the GAM adjusted coefficient of determination was greater than 0.10" for more clarity.

Results

L212f: This is already a discussion of your result and should thus be part of the discussion section.

We agree the sentence should be moved to the discussion in section 4.1.

L231f: Check this sentence. Better "fitted to XX DOC concentration time series"?

We agree with the suggestion to modify the sentence as : "Of the 185 catchments, GAMs were fitted for 159 to DOC concentrations time series, 168 to NO3 concentrations time series, 162 to SRP concentrations time series, and 185 to discharge time series".

L241: Check this sentence. Discharge cannot have a seasonal concentration cycle.

We suggest rephrasing the sentence as: "Most of the catchments had a seasonal concentration cycle: 85%, 71%, 78%, for NO3, DOC, SRP concentration respectively and 100% of them had a seasonal discharge cycle".

L244: Does that refer to the comparison between all catchments? That is not clear here.

Yes it does. We suggest rephrasing as: "The annual phases for discharge were more stable among all catchments than those for concentrations".

L245f: I am not sure were to see this gradient in Fig. 4. Is that referring to the right figure?

Yes, we should specify that this is referring to Fig. 4d (and Supplemental S7) which shows that the relative amplitude of discharge seasonal variations are more or less important depending on the catchments.
L257f: What does that stability means? That the pattern does not change between the years? This cannot be seen from the GAM averaging over all years. I am a bit lost here.

It means that these two metrics are stable between all catchments. Indeed, we suggest clarifying by rephrasing: "The DOC MaxPhase and NO3MinPhase were the same for all catchments as they always occurred between July and December (Fig.4, Supplemental S7).

L288: You may give direction of the correlation with the hydrologic variables as well.

We suggest rephrasing as: "It correlated most strongly with soil P stock (rs=-0.40), climate and hydrology (rs=-0.43 to -0.34 with effective rainfall, Qmean, QMNA), elevation, and hydrographic network density".

Discussion

L304ff: Rather than directly with the interaction of N and C wouldn't it be better to first explain the individual spatial patterns?

Because the individual patterns of NO3 and DOC are opposite, we think that it makes sense to explain them together. We argue that the quality of this discussion section was highlighted by referee #2 and that individual interpretations of DOC and NO3 would lead to redundant paragraphs. Therefore, we think this is worth to keep this structure for the discussion section.

L313ff: But this argument would lead to high concentrations of both, C and N?

If high SOC content in such soils are associated to higher N leaching this lead to a reservoir rich in organic Carbon but poor in Nitrogen.

L324ff: Wouldn't Fovet et al. (2018, 10.1016/j.jhydrol.2018.02.040) provide a good mechanistical backup for the processes described here?

Indeed, similar mechanisms of mixing lateral (along the hillslopes) and vertical (with

depth) gradients of elements sources are discussed in Fovet et al. (2018) but for interpreting temporal patterns observed during rainfall-discharge events. We agree with the recommendation of referee#1 to illustrate the interpretation of temporal patterns using a conceptual diagram (see reply to major comment 3 above).

L334ff: You need some references for these statements.

We suggest adding the following references: Davidson et al., (2006); Hénault and Germon, (2000); Luo and Zhou, (2006)

Davidson, E. A., Janssens, I. A., and Luo, Y.: On the variability of respiration in terrestrial ecosystems: moving beyond Q10, Global Change Biology, 12, 154-164, 2006.

Hénault, C. and Germon, J. C.: NEMIS, a predictive model of denitrification on the field scale, European Journal of Soil Science, 51, 257-270, 2000.

Luo, Y. and Zhou, X.: CHAPTER 5 - Controlling Factors. In: Soil Respiration and the Environment, Luo, Y. and Zhou, X. (Eds.), Academic Press, Burlington, 2006.

L400f: You may show and quantify this earlier on by the ratio of CVc and CVq as done in Thompson et al. (2011, 10.1029/2010wr009605).

Many recent papers on the temporal variability in C and Q have used the CV ratio as a descriptive metrics. We decided to use different metrics here, specifically focusing on seasonality is an originality of our analysis compared to published work of others.

SI Fig. S1: Panel b does not make sense without a legend. Typo in panel d legend name.

Figure S1 has been corrected.
* * *
Table R1: Spearman's rank correlations between the C10, C50 and C90 metrics for each element

|  | DOC | | NO3 | | SRP | |
|---|---|---|---|---|---|---|
|  | C50 | C90 | C50 | C90 | C50 | C90 |
| C10 | 0.89 | 0.56 | 0.87 | 0.56 | 0.9 | 0.78 |
| C50 |  | 0.71 |  | 0.83 |  | 0.93 |

**Fig. 1.** Table R1: Spearman's rank correlations between the C10, C50 and C90 metrics for each element

[Figure]

**Fig. 2.** Figure R1 :Correlation matrix between Headwater catchment descriptors, Spearman coefficients are visible when p-value > 0.05.

[Figure]

**Fig. 3.** Figure R2 (new Figure 8) : Conceptual diagram of seasonal flowpaths involved in the DOC-NO3 seasonal cycles leading to a) in-phase cycles with discharge or b) out-of-phase cycles with discharge.

(a)

Effective rainfall [mm]
500
700
900
1100

(b)

Major geology classes
Granite and gneiss
Schist and micaschist
Other (sedimentary rocks…)

(c)

Type of areas
Urban
Agricultural
Semi-natural
Natural

(d)

Point sources
Domestic
Industrial

(e)

N surplus [kgN/ha]

(f)

P surplus [kgP/ha]

**Fig. 4.** Figure S1 corrected

---

## Author Comment (AC2) · 21 Oct 2020

We thank Referee #2 for his/her positive evaluation of the study: "The multi-element, many sight approach utilized does provide interesting insight into the potential influences of changing seasonal hydrology/ flowpath and landscape characteristics on the biogeochemistry of the study region."

He/she raised two major comments:

1) "The paucity of other studies focusing on multi-element patterns, in headwater streams, that examine seasonal patterns, or that focus on multiple catchments is some-

[Figure]

Creative Commons BY license logo

what overemphasized in the framing of the research though and further cross comparison with studies that include all or only some of those criteria would benefit the introduction and discussion."

And specific comment on Lines 65-75- "I understand the point that the authors are making here, but there are actually a number of studies meeting most of these criteria that could be helpful in interpretation of results and in understanding the generality of the patterns observed across regions. A couple of ideas that came to mind when reading this section were:

-Fasching et al. 2019 in Ecosystems also use GAM models and the approach used to explore multiple drivers may be helpful, Natural land cover in agricultural catchments alters flood effects on DOM composition and decreases nutrient levels in streams - https://doi.org/10.1007/s10021-019-00354-0

-Although larger watersheds in the region are also included in the analysis I would suggest that some comparison should be made with Moatar et al. 2017, WRR, Elemental properties, hydrology, and biology interact to shape concentration- discharge curves for carbon, nutrients, sediment, and major ions https://doi.org/10.1002/2016WR019635

-The review and conceptual paper presented by Kaushal et al. 2018 in Biogeochemistry may also be helpful in evaluating the role of season and land use on multi-element water chemistry."

Indeed, and in the Introduction paragraph L45-54, factors of spatial variability in concentrations are reviewed from various contexts (headwaters or not) and from studies that analyzed at least one of the three elements. Similarly, the following paragraph L. 55-65 reviews seasonal variations in at least of the three element concentrations but without filter on catchment size or number of catchments included in the analysis. Therefore, we highlighted the scarcity of studies dealing with multi-element and multiple catchments, in headwater streams and including analysis of seasonal pattern in the introduction section only to describe the need for more investigation, which our work

aims to contribute to (Lines 66-74).

We thank referee # 2 for the relevant additional references, and according recommendations from referee # 1 too, we suggest the following modifications in order to position our study in regards to these published results in the introduction:

"Besides being spatially variable, C, N, and P concentrations also vary temporally. The variability of concentrations with flow has been described in several studies using concentration-flow relationships at event (Fasching et al., 2019) or inter-annual to long-term scales (Basu et al., 2010; 2011; Moatar et al., 2017). Concentrations also vary seasonally in streams and rivers ..."

"We hypothesized that: 1) Human (i.e. rural and urban) pressures determine spatial variability in NO3 and SRP concentrations (Preston et al., 2011; Melland et al., 2012; Dupas et al., 2015; Kaushal et al., 2018), while soil and climate characteristics determine that in DOC and possibly SRP (Lambert et al., 2011; Humbert et al., 2015; Gu et al., 2017)."

Please see also the reply to referee # 1, major comment 2.

2) "Regarding the GAM model used to describe seasonality, this is a useful approach, but I also wonder if there may be opportunity to modify the presentation and possibly the models slightly to explore interactions between multiple drivers (e.g. season x land use or flow x soil)."

We thanks referee #2 for the suggested reference of Fasching et al., 2019, which is indeed very relevant here. In the presented study, we used GAM to described the seasonal patterns from concentration measurements. We used then correlation analyses with Land uses, flow and soils to see if they had a relationship or not with those seasonal patterns. The approach suggested by referee # 2 to fit the GAM according to time but also land use, flow and soils could be another way to explore these relationships indeed but the possible interpretation of the GAM should not be different from the one

we could have using the correlation analysis.

Note also that, we tested a GAM fitting using both the month and the year in order to extract a long-term component (lines 175-179). The model sometimes failed in converging, and then it seems reasonable to limit the GAM complexity and to keep a two-steps analysis: 1) extracting seasonality using GAM and 2) analyzing the relationships between the extracted seasonality and the geographical variables.

Reply to specific comments

Lines 45-50 – "There have been a number of studies in Canada and United States to evaluate the influence of agricultural land use on DOC concentration and DOM composition. Although the statement that composition is usually quite altered is true, often concentration is more a function of the same factors as in non-agricultural catchments, in particular the presence of wetlands and soil drainage properties."

Indeed, DOC concentration has been primarily linked to topography and presence of wetlands and saturated areas which is true both in forested and agricultural catchments. As also suggested by referee #1, we suggest adding more references (lines 45-47):

"DOC concentration in streams has been related to topography, wetland coverage, and soil properties such as clay content or pH (Andersson and Nyberg, 2008; Brooks et al., 1999; Creed et al., 2008; Hytteborn et al., 2015; Temnerud and Bishop, 2005; Zarnetske et al., 2018; Musolff et al., 2018)."

Line 68 - This is true, but there is a lot of study that goes on further upstream in even smaller catchments where land management can be linked directly to impact.

Indeed, we did not state that there were no literature at the scale of headwater catchments: several studies at such scales in agricultural or impacted contexts focused on the link between specific land management practices and water quality. However, such studies rarely compare more than 100 catchments like we did in the present study in

order explore the spatial variability of this link between land management and impacts.

Line 73- maybe also ad "multi-element" to this statement because there are many studies that examine multi-catchment patterns for a single element.

We suggest to rephrase as "multiple-catchment studies" on multiple elements are uncommon".

Line 109- This is good. Often selecting sites in a stream network without spatial independence is a pitfall for many site studies in a region, particularly when working with data where the authors did not chose the original sampling locations.

Yes, it was for us an important criterion to focus the analysis on the spatial variability and not on the "longitudinal" variability within nested catchments.

Line 111- Please explain why these criteria were used for outlier selection and how commonly extremely high concentrations were observed.

The concentration databases initially included some extremely high maximum NO3, PO4 and Ptot values. We could clearly interpret these as outliers. Our thresholds for the selection of outliers (values > 200 mg N.L-1 or 5 g P.L-1) were chosen: 1) by expert advice (producer of the data) and 2) after verification on the data (in terms of proportions of values eliminated on each time series and number of time series concerned). Among the 185 NO3 time series, 3 were concerned and for Phosphorus 5 were concerned. Only one value was removed by time series.

109-112 – Were data examined to ensure that there were not seasonal biases in the timing of missing data and that certain sites were not heavily sampled only in one season (summer samples only for example)

We have imposed a criterion for selecting the time series according to the sampling frequency (at least 6 years of data with at least 8 values per year). We also looked at the data to see which months were least sampled and in the OSUR database no bias was observed as it is based on fixed and regular frequencies while in the HYDRE /

BEA we noticed a few time series where summer periods were actually less sampled but for some years only(over the 10 years). We suggest adding this information in the main text.

Line 185- The seasonality metric is interesting, but doesn't really separate the flow condition or discharge from other factors like temperature that vary seasonally. Calculation of a similar metric for high flow vs low flow for comparison to the SI might be quite revealing. An example of that method is in Fasching et al. 2019.

Indeed, but in the studied catchments, high flows are well in phase for all the catchments with maximum of discharge in winter (colder season) and low flows are all occurring at the end of summer (warmer season). Therefore, the suggested metric is relevant but it would lead to the same results as our seasonal index with this data set of catchments. However, a seasonal index based on season only has the advantage of being applicable even if there is no stream flow data, and in such case, the interpretation of the index should be adapted of course.

Figure 4 – I think the information displayed here is valuable, but I wonder if a visual with additional information might be possible with the GAM results if the influence of 2 different drivers were displayed in a 3d version of the figure similar to Figure 7 in Fasching et al. 2019. It could be discharge or land use on the other axis.-

We think that the use of the GAM proposed by Fasching et al., 2019 is fully valuable and interesting. However, in the way we used the GAM here, we first smooth the observations to compute metrics on the average seasonal pattern of concentrations, and then, we investigated potential drivers within a correlation analysis between catchment descriptors and concentration metrics. Again, given the relative moderate number of concentration points in each station, fitting the GAM on both temporal (month) and spatial (geographic variables such as discharge or land uses) variables could be difficult (see also reply to major comment 2).

Discussion-The discussion on DOC/NO3 patterns is well written and I agree with the

authors general interpretation of the results.

Thank you.

For the SRP discussion it may be worthwhile to reference the strong correlations that have been observed in small agricultural catchments between soil P and runoff concentrations. There are metrics included in the predictor dataset for TP_soil and P surplus which appear to be model outputs. It may help with interpretation of results if it can be noted whether these follow anticipated patterns of buildup where more intensive livestock or fertilizer input is occurring.

We suggest adding such discussion to subsection 4.3., line 376:

"Nonpoint sources of P in agricultural runoff, historical inputs of fertilizer and manure in excess of crop requirements have led to a build up of soil P levels, particularly in areas of intensive crop and livestock production (Sharpley et al., 1994). This led to correlations between soil P and runoff concentrations in agricultural catchments (Cooper et al., 2015; Sandström et al., 2020), as found here."

Sharpley, A. N., et al. (1994). "Managing Agricultural Phosphorus for Protection of Surface Waters: Issues and Options." Journal of Environmental Quality 23(3): 437-451.

Sandström, S., et al. (2020). "Particulate phosphorus and suspended solids losses from small agricultural catchments: Links to stream and catchment characteristics." Science of The Total Environment 711: 134616.

Line 380 – In the context of the observed seasonal pattern can you comment on the timing of nutrient applications and whether there is potential for depletion of soluble sources over time or not.

As explained in reply to previous comment, the inputs of fertilizer and manure in excess of crop requirements have led to a buildâ ĂŘup of soil P legacy storage (Sharpley et al., 1994), which gradually leaches into the water for decades (Sandström et al., 2020).

Therefore, the timing of current nutrient applications is likely to be invisible in the stream concentrations due to such time lags. Therefore, the correlations found between SRP C50 and variables related to P sources (TP_soil, domestic point sources, P surplus. . .) are significant but weaker (Line 287).

Table 1 – Presumably some fields are used for both summer and winter crops. A total % cropland variable might be useful if not already considered

The "Winter crop" variable corresponds to crops with a winter plant cover and a phenological maximum in April, thus relating to three major crops: wheat, barley and rapeseed. The "Summer crop" variable corresponds to crops with bare winter soil and a phenological maximum in early summer (July), thus relating to two major crops: corn (and sunflower but it is not cultivated in the studied region). We distinguished these two types in order to refine the proxy of pressures regarding potential NO3 leaching (higher for summer crops because of potentially bare winter soils). Adding the total percentage of cropland would not add more information than the percentages of grassland and forest.

---

## Author Response (ED1)

**2nd revision - Reply to Anonymous Referee #1**

The response to the general comment is developed in the specific questions, particularly 3 and 6.

1. *L55: Homogenization of formerly (naturally) heterogenerous catchments is argued to be the main reason for nitrate chemostatic behavior and thus purely transport-limited nitrate exports. This does not fit to your description of homogeneity as a characteristic of natural catchments.*

The generalization of the drivers from C to C, N and P is mainly due to human pressure which could induce internal storage, and therefore, purely transport-limited nitrate exports. We stress now on pressure, rather than on homogeneity, for such generalization.

2. *L66 and L87: You boil down seasonality to a hydrological control (controlling other biogeochemical processes). What about light and temperature. In-stream ecosystems but also shallow near stream systems may be also be seasonally controlled by that.*

Light and temperature are now mentioned. Investigations on the effect of light and temperature on C, N and P concentrations are poor, because they really vary along the stream. Temperature does not vary a lot over Brittany, and light cannot easily be captured by simple indicators.

3. *L103: This sentence does not make sense for me.*
Line suppressed.

4. *L181ff: For me this is still not resolved and need to be explicitly mentioned here. By GAM you cannot analyze catchments without seasonality. Your SI metric thus does not cover the whole range of analysed catchments. If you want it like that, its ok. But you need to mention it.*
A sentence added. "GAM provides SI metrics on seasonality, which can be easily linked with different geographical variables. However, it cannot take into account catchments without seasonality."

5. *Table 1: You added the information on the Topo_i. I just don't see the point why you don't use the established name "topographic wetness index" here. Moreover it is not clear how you derive one value for the entire catchment. Usually the Beven-Kirkby approach derives this for each grid cell of the DEM. Finally, the unit seems to be wrong: log[ha]? Note that Beven and Kirkby use the local slope in radian not %.*
Corrected. Indeed, it is preferable to use the name "average topographic wetness index" (meanTWI) rather than "downstream topographic index" (Topo_i), even if the usual TWI in the literature takes into account the local slope rather than the downstream slope.
We derived the value for each cell of the DEM grid according to the Beven-Kirkby approach:

$TWI = log \frac{\alpha}{\tan \beta}$. Where $\alpha$ is the drainage area (ha) and $\beta$ is the downstream slope (%) (Merot et al., 2003). Then we averaged the values of each cell present on the drained surface to obtain a "meanTWI" value for the entire watershed, expressed in log(Ha).

6. *L205: Again I ask for a bit more words on the idea of the PCA. Tell the reader why you use this technique, what it can tell you and what are potential drawbacks that have to be kept in mind. For instance, PCA is a linear method and explained variance may be misleading when relations between the elements are non-linear (e.g. the power law that Taylor and Townsend propose for C:N). Your rank correlation acknowledge that but this, again is not mentioned. It would help the paper elaborating on this to not leave the impression you randomly chose you methods which surely is not the case.*

A sentence added. "PCA was chosen, despite it assumes linear relationships between variables, because it provides a graphical representation of correlations between variables or groups of variables and their contributions the variance."

7. *L233: I criticized the use of PCA in the first review. Now you moved it to the Supplement which does not solve the issue but rather complicates it. If you want to use it, it's ok but needs to be justified (see above comment) and then properly used in the main manuscript.*

See the sentence added L205. Add the results of PCA include again in the main manuscript.

8. *L237: This link to S4b is for C50 only - should be mentioned here. Wouldn't it make sense and be easier to have a correlation matrix for all concentration metrics across all constituents in the SI?*

It's done. We added the precision on C50 and the S8 supplement reference (correlation matrix).

9. *L242: Here you use SI the first time in the results. At this point it would be good to mention that this does not incorporate all catchments and give the number of cases?*

A sentence was added "…, calculated on the catchments for which a GAM can be fitted, i.e. presenting a seasonal feature,… ".

10. *L295f: If the classification of "in-phase" and "out-of-phase" refers to van Meter et al. 2019, it would be fair to mention it.*

Reference added.

11. *L414: The meaning of a synchronization of a contribution with maximum flow is not fully clear to me.*

Definition added.

12. *L415: Redox-processes may be mentioned explicitly as this seems solely refer to assimilation.*

"Redox-processes" added.

13. *L422: I know what you mean but expression is not fully transporting that. Maybe: "decrease NO3 concentration more in shallower and younger groundwater than in deeper, older groundwater"?*

Corrected.

14. *L495ff: Without explicitly mentioning it the entire monitoring and management implications are boiled down to the discussion of C:N ratios which haven't been presented in the results at all. and actually no implications are mentioned in this section. I suggest to broaden the view here a bit and share with the reader how your result may influence water quality management and monitoring design.*

Few sentences on consequences on water quality management and monitoring design have been added.

"However, we can stress that monitoring C-N and P is important as each of these elements can follow a specific pattern, even in neighboring catchments. Yet, these three basic elements are not always analyzed in water quality monitoring. Therefore, sample points for which monitoring associate these three elements have to be preserve for a long term monitoring. They will be necessary to further investigate their variations in relation with geomorphological and climate conditions. In this paper, we used inter-annual mean values for DOC, $NO_3$ and SRP loads to establish the spatial variability and the seasonal patterns across headwater catchments. Because we demonstrated that seasonality index (SI) and flow flashiness ($W_2$) are linked, our results can be used to classify non-monitored catchments as a function of their potential load flashiness. Flow flashiness ($W_2$) combined with SI or the slope of C-Q relationships for high flows, could be employed for a sampling monitoring design to improve annual or seasonal load estimations for the most contributive catchments (Moatar et al, 2020). Yet, other issues, such as the assessment of eutrophication risk for some lakes, estuaries or bays around the peninsula would require more frequent sampling especially for SRP. "

*The authors have done a thorough job of addressing reviewer comments and revising this paper.*

*I appreciate the clarification provided on statistical methods and the potential challenges to implementing an expanded GAM model. As the authors suggest in their responses, the approach to statistical analysis being altered is unlikely to change the over conclusions of the descriptive study. Suggestions of an expanded GAM model based on reading the first draft of the manuscript were mainly out of interest as to whether an alternative statistical approach would be complementary in simplifying or integrating presentation of some of the results and exploring interactions between geographic variables, hydrology, and concentration. As currently presented the results clearly highlight seasonality very well. Also, where discharge and concentrations follow similar seasonality it is visually evident through the GAM results. However, I do wonder if there might still be some room to explore whether the geographic variables influencing seasonality of discharge are the same as those for concentrations. This could just be a simple addition of correlations between discharge seasonality metrics and geographic variables to Table 3. That would allow for a test of whether the controls are similar*

As suggested, correlations between discharge seasonality metric and geographic variables have been added to Table 3. And comments introduced in lines 376-385.

[revised manuscript text omitted]

35    **Figure S~4~5. Density histogram of variance in the seasonal component explained by the Generalized Additive Models (GAMs) among headwater catchments for dissolved organic carbon (DOC), nitrate (NO₃), soluble reactive phosphorus (noted PO4 here), and discharge (Q). Rsq is the coefficient of determination between observed concentrations and values calculated by the GAM. Dashed lines identify mean values.**

[Figure]

**(1)**

[Figure]

**(2)**

Figure S6̶5̲. Two examples of Generalized Additive Model adjustments to nitrate (N-NO₃) time series: (1) La Loisance River (station
45    no. 04162958 in the OSUR database) illustrates poor adjustment quality (Rsq=0.02). (2) Saint Niel River/ tributary of the Blavet
(station no. 04191980 in the OSUR database) illustrates good adjustment quality (Rsq=0.66).

[Figure]

**Figure S6. On average, the Generalized Additive Model (GAM) seasonal components were fitted to time series of monthly data,
which is a low frequency for investigating intra-annual variations. We assumed that aggregating the 10 years would allow a relatively
robust average seasonal pattern to be extracted. To verify this assumption, we analyzed how much the relatively low-frequency
sampling influenced the seasonal metrics. We calculated differences between the seasonal indices calculated from GAMs adjusted
to high-frequency data and those calculated from GAMs adjusted to monthly data, generated by random Monte Carlo draws
(n=1000) from the high-frequency time series. This analysis was performed for three catchments Brittany for which $NO_3$ and SRP
concentration data were available at higher frequency (not available for DOC data) from 2007-2016. The figure summarizes the
variability in the seasonality indices (Ampli, seasonality index (SI), MaxPhase and MinPhase) calculated from the 1000 GAM models
that were fitted to monthly concentration time series for (a) nitrate ($NO_3$) in the Kervidy-Naizin catchment, (b) $NO_3$ in the Néal
catchment, and (c) soluble reactive phosphorus (SRP) in the Le Queffeuth catchment. Only significant GAM models (Rsq ≥ 0.10)
are shown. Dashed lines indicate the value obtained by the GAM with the original daily (Kervidy-Naizin, AgrHyS observatory) or
weekly (Néal and Le Queffeuth catchments) time series. The y-axis corresponds to the number of catchments (N). The comparisons
show that the distribution of seasonality metrics obtained from lower-frequency time series were centered on the values obtained
from the original time series. Despite some delay in phases, minimum and maximum concentrations were identified during the same
season by both types of time series. For $NO_3$, the mean errors in seasonal metrics obtained from the monthly time series were -4.5%
and 6.7% for amplitude, 21.0% and -7.2% for SI for the Kervidy-Naizin and Néal catchments respectively. PhaseMax was delayed
by -1.0 to -1.5 months, and PhaseMin by 0 to 1 months. For SRP (Le Queffeuth catchment), the mean error was -4.0% for amplitude,
-7.0% for SI, ±18 days for PhaseMax, and ±12 days for PhaseMin.**

[Figure]

**Figure S7. Histograms of water quality metrics and discharge, from left to right: absolute annual amplitude (mg.l⁻¹), annual minimum and maximum concentration phases (in months), and the seasonal index (only for concentrations). From top to bottom: for dissolved organic carbon (DOC) (n=113), nitrate (N-NO₃) (n=142), soluble reactive phosphorus (SRP) (n= 126), and discharge (Q) (n=124). Non-significant amplitudes (Rsq<0.1) are not shown.**

[Figure]

**Figure S8. Matrices of Spearman's rank correlations between concentration percentiles (10$^{th}$ (C10), 50$^{th}$ (C50), and 90$^{th}$ (C90)) and amplitudes of dissolved organic carbon (DOC), nitrate N (noted here NO$_3$), and soluble reactive phosphorus (noted here PO4). Only significant (p ≤ 0.05) values are shown.**